# LOCALITY-AWARE GRAPH REWIRING IN GNNS

**Federico Barbero**[1],[*] **Ameya Velingker**[2], **Amin Saberi**[3], **Michael Bronstein**[1], **Francesco Di Giovanni**[1]
[1]University of Oxford, Department of Computer Science
[2]Google Research
[3]Stanford University, Department of Management Science and Engineering

## ABSTRACT

Graph Neural Networks (GNNs) are popular models for machine learning on graphs that typically follow the message-passing paradigm, whereby the feature of a node is updated recursively upon aggregating information over its neighbors. While exchanging messages over the input graph endows GNNs with a strong inductive bias, it can also make GNNs susceptible to *over-squashing*, thereby preventing them from capturing long-range interactions in the given graph. To rectify this issue, *graph rewiring* techniques have been proposed as a means of improving information flow by altering the graph connectivity. In this work, we identify three desiderata for graph-rewiring: (i) reduce over-squashing, (ii) respect the locality of the graph, and (iii) preserve the sparsity of the graph. We highlight fundamental trade-offs that occur between *spatial* and *spectral* rewiring techniques; while the former often satisfy (i) and (ii) but not (iii), the latter generally satisfy (i) and (iii) at the expense of (ii). We propose a novel rewiring framework that satisfies all of (i)–(iii) through a locality-aware sequence of rewiring operations. We then discuss a specific instance of such rewiring framework and validate its effectiveness on several real-world benchmarks, showing that it either matches or significantly outperforms existing rewiring approaches.

## 1 INTRODUCTION

Graph Neural Networks (GNNs) (Sperduti, 1993; Goller & Kuchler, 1996; Gori et al., 2005; Scarselli et al., 2008; Bruna et al., 2014; Defferrard et al., 2016) are widely popular types of neural networks operating over graphs. The majority of GNN architectures act by locally propagating information across adjacent nodes of the graph and are referred to as Message Passing Neural Networks (MPNNs) (Gilmer et al., 2017). Since MPNNs aggregate messages over the neighbors of each node recursively at each layer, a sufficient number of layers is required for distant nodes to interact through message passing (Barceló et al., 2019). In general, this could lead to an explosion of information that needs to be summarized into fixed-size vectors, when the receptive field of a node grows too quickly due to the underlying graph topology. This phenomenon is known as **over-squashing** (Alon & Yahav, 2021), and it has been proved to be heavily related to topological properties of the input graph such as curvature (Topping et al., 2022) and effective resistance (Black et al., 2023; Di Giovanni et al., 2023).

Since over-squashing is a limitation of the message-passing paradigm that originates in the topology of the input-graph, a solution to these problems is **graph rewiring** (Topping et al., 2022), in which one alters the connectivity of the graph to favor the propagation of information among poorly connected nodes. *Spatial rewiring* techniques often connect each node to any other node in its $k$-hop (Brüel-Gabrielsson et al., 2022; Abboud et al., 2022), or in the extreme case operate over a fully-connected graph weighted by attention – such as for Graph-Transformers (Kreuzer et al., 2021; Mialon et al., 2021; Ying et al., 2021; Rampasek et al., 2022). *Spectral rewiring* techniques instead aim to improve the connectivity of the graph by optimizing for graph-theoretic quantities related to its expansion properties such as the spectral gap, commute time, or effective resistance (Arnaiz-Rodríguez et al., 2022; Karhadkar et al., 2022; Black et al., 2023).

While graph rewiring is a promising direction, it also introduces a fundamental trade-off between the preservation of the original topology and the 'friendliness' of the graph to message passing. Spatial rewiring techniques partly preserve the graph-distance information (i.e. its '*locality*') by

---

[*]Correspondence to federico.barbero@cs.ox.ac.uk.

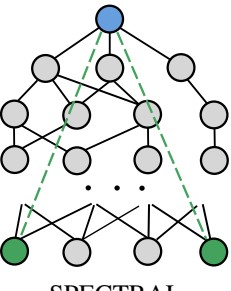 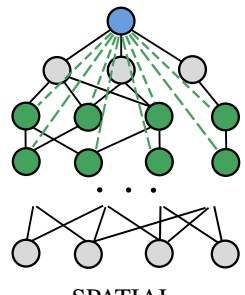 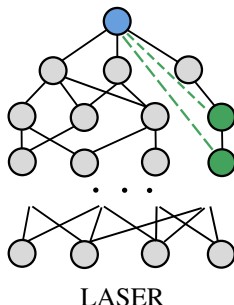

SPECTRAL                    SPATIAL                    LASER

Figure 1: Difference between spectral (left), spatial (middle), and **LASER** (right) rewirings in green with respect to the blue node of reference. Spectral rewirings are sparse and connect distant nodes. Spatial rewirings are able to retain local inductive biases at the cost of sparsity. **LASER** remains both local and sparse by optimizing over the edges to be added.

only adding edges within a certain radius or by relying on positional information. However, these methods often result in a dense computational graph that increases memory complexity and can cause issues such as over-smoothing (Nt & Maehara, 2019; Oono & Suzuki, 2020; Rusch & Mishra, 2020; Di Giovanni et al., 2022). Conversely, spectral rewiring approaches add fewer edges according to some optimization criterion and hence better preserve the sparsity of the input graph. However, these methods 'maximally' destroy the locality induced by the graph since they typically insert very 'long' edges among distant nodes (see Figure 1). The following natural question then arises: *Can we design a general graph rewiring framework that leverages the inductive bias of spatial methods but in a more edge-efficient way characteristic of spectral methods?*

**Contributions and outline.**   In this work, we address the above question by proposing a general framework for graph-rewiring that improves the connectivity, while preserving locality and sparsity:

- In Section 3 we review existing rewiring approaches and classify them as either *spatial* or *spectral*, highlighting their limitations. We then provide a general list of desiderata for rewiring that amounts to (i) reducing over-squashing, and preserving both (ii) the graph-locality and (iii) its sparsity.

- In Section 4 we introduce a *paradigm* for rewiring that depends on arbitrary connectivity and locality measures. We argue that in order to satisfy (i)–(iii) above, a single rewiring is not enough, and instead propose *sequential rewiring*, where multiple graph snapshots are considered. Building on Karhadkar et al. (2022), we also draw an important equivalence between graph-rewiring on one side, and multi-relational GNNs and temporal-GNNs on the other.

- In Section 5 we present a specific instance of the aforementioned paradigm termed **L**ocality-**A**ware **SE**quential **R**ewiring (**LASER**). Our framework leverages the distance similarly to spatial rewiring while also guaranteeing the efficiency of spectral techniques by sampling edges to add according to equivariant, optimal conditions. We show that **LASER** reduces over-squashing and better preserves the locality of the graph compared to spectral rewiring techniques.

- In Section 6 we validate **LASER** on different tasks, attaining performance that is on par or superior to existing rewiring techniques. In particular, we present extensive ablation studies to support our claim that **LASER** is more efficient than spatial methods while being better at preserving graph-distance information in comparison to spectral approaches.

## 2 BACKGROUND

**Preliminaries on graphs.**   Let $\mathsf{G} = (\mathsf{V}, \mathsf{E})$ be an undirected graph with $n$ nodes $\mathsf{V}$ and edges $\mathsf{E}$, which are encoded by the non-zero entries of the adjacency matrix $\mathbf{A} \in \mathbb{R}^{n \times n}$. Let $\mathbf{D}$ be the diagonal degree matrix such that $\mathbf{D}_{vv} = d_v$. We recall that the normalized graph Laplacian $\mathbf{\Delta} = \mathbf{D}^{-1/2}(\mathbf{D} - \mathbf{A})\mathbf{D}^{-1/2}$ is a symmetric positive semi-definite operator with eigenvalues $0 = \lambda_0 \leq \lambda_1 \leq \cdots \leq \lambda_{n-1}$. We assume that $\mathsf{G}$ is connected, so that $\lambda_1 > 0$ and refer to it as the *spectral gap*. From the Cheeger inequality, it follows that a larger $\lambda_1$ generally means better connectivity of $\mathsf{G}$. We denote by $d_{\mathsf{G}}(u, v)$ the shortest-path distance between the nodes $u, v$. We finally recall that a random walk on $\mathsf{G}$ is a Markov chain on $\mathsf{V}$ with transition matrix $\mathbf{D}^{-1}\mathbf{A}$ and that the **commute time** CT is defined as the expected number of steps required for a random walk to commute between

two nodes. Note that the commute time $\mathsf{CT}(v, u)$ between two nodes $v$ and $u$ is proportional to their **effective resistance** $\mathsf{R}(v, u)$ (Chandra et al., 1996) as $\mathsf{CT}(v, u) = 2|\mathsf{E}|\mathsf{R}(v, u)$.

**The message-passing paradigm.** We consider the case where each node $v$ has a feature $\mathbf{x}_v^{(0)} \in \mathbb{R}^d$. It is common to stack the node features into a matrix $\mathbf{X}^{(0)} \in \mathbb{R}^{n \times d}$ consistently with the ordering of $\mathbf{A}$. GNNs are functions defined on the featured graph that can output node, edge, or graph-level values. The most common family of GNN architectures are Message Passing Neural Networks (MPNN), which compute latent node representations by stacking $T$ layers of the form:

$$\mathbf{x}_v^{(t)} = \mathsf{up}^{(t)}(\mathbf{x}_v^{(t-1)}, \mathsf{a}^{(t)}(\{\!\!\{\mathbf{x}_u^{(t-1)} : (v, u) \in \mathsf{E}\}\!\!\})), \tag{1}$$

for $t = 1, \ldots, T$, where $\mathsf{a}^{(t)}$ is some permutation-invariant *aggregation* function, while $\mathsf{up}^{(t)}$ *updates* the node's current state with aggregated messages from its neighbors.

**Over-squashing and long-range interactions.** While the message-passing paradigm usually constitutes a strong inductive bias, it is problematic for capturing long-range interactions due to a phenomenon known as *over-squashing*. Given two nodes $u, v$ at distance $d_\mathsf{G}(u, v) = r$, an MPNN will require $T \geq r$ layers to exchange messages between them. When the receptive fields of the nodes expand too quickly (due to volume growth properties characteristic of many real-world scale free graphs), the MPNN needs to aggregate a large number of messages into fixed-size vectors, leading to some corruption of the information (Alon & Yahav, 2021). This effect on the propagation of information has been related to the Jacobian of node features decaying exponentially with $r$ (Topping et al., 2022). More recently, it was shown that the Jacobian is affected by topological properties such as effective resistance (Black et al., 2023; Di Giovanni et al., 2023).

## 3 EXISTING GRAPH-REWIRING APPROACHES AND THEIR LIMITATIONS

The main principle behind graph rewiring in GNNs is to decouple the input graph $\mathsf{G}$ from the computational one. Namely, *rewiring* consists of applying an operation $\mathcal{R}$ to $\mathsf{G} = (\mathsf{V}, \mathsf{E})$, thereby producing a new graph $\mathcal{R}(\mathsf{G}) = (\mathsf{V}, \mathcal{R}(\mathsf{E}))$ on the same vertices but with altered connectivity. We begin by generalizing the MPNN formalism to account for the rewiring operation $\mathcal{R}$ as follows:

$$\mathbf{x}_v^{(t)} = \mathsf{up}^{(t)}(\mathbf{x}_v^{(t-1)}, \mathsf{a}_\mathsf{G}^{(t)}(\{\!\!\{\mathbf{x}_u^{(t-1)} : (v, u) \in \mathsf{E}\}\!\!\}), \mathsf{a}_{\mathcal{R}(\mathsf{G})}^{(t)}(\{\!\!\{\mathbf{x}_u^{(t-1)} : (v, u) \in \mathcal{R}(\mathsf{E})\}\!\!\})), \tag{2}$$

where a node feature is now updated based on information collected over the input graph $\mathsf{G}$ and the rewired one $\mathcal{R}(\mathsf{G})$, through (potentially) independent aggregation maps. Many rewiring-based GNN models simply exchange messages over $\mathcal{R}(\mathsf{G})$, i.e., they take $\mathsf{a}_\mathsf{G} = 0$. The idea of rewiring the graph is implicit to many GNNs, from using Cayley graphs (Deac et al., 2022), to virtual nodes (Cai et al., 2023) and cellular complexes (Bodnar et al., 2021). Other works have studied the implications of *directly* changing the connectivity of the graph to de-noise it (Klicpera et al., 2019), or to explore multi-hop aggregations (Abu-El-Haija et al., 2019; Ma et al., 2020; Wang et al., 2020; Nikolentzos et al., 2020). Ever since over-squashing was identified as an issue in MPNNs (Alon & Yahav, 2021), several novel rewiring approaches have been proposed to mitigate this phenomenon.

**Related work on spatial rewiring.** Most spatial rewiring models attempt to alleviate over-squashing by adding direct connections between a node and every other node within a certain distance (Brüel-Gabrielsson et al., 2022; Abboud et al., 2022) — with (dense) Graph Transformers being the extreme case (Ying et al., 2021; Mialon et al., 2021; Kreuzer et al., 2021; Rampasek et al., 2022). These frameworks follow equation 2, where $\mathsf{a}_\mathsf{G}$ and $\mathsf{a}_{\mathcal{R}(\mathsf{G})}$ are learned independently, or the former is zero while the second implements attention over a dense graph. Spatial rewiring reduces over-squashing by creating new paths in the graph, thus decreasing its diameter or pairwise effective resistances between nodes. The rewired graph still preserves some information afforded by the original topology in the form of distance-aware aggregations in multi-hop GNNs, or positional encoding in Graph-Transformers. A drawback of this approach, however, is that we end up compromising the sparsity of the graph, thereby impacting efficiency. Thus, a natural question is whether *some of these new connections introduced by spatial rewiring methods may be removed without affecting the improved connectivity.*

We also mention spatial rewiring methods based on improving the curvature of $\mathsf{G}$ by only adding edges among nodes at distance at most two (Topping et al., 2022; Nguyen et al., 2022). Accordingly,

these models may fail to significantly improve the effective resistance of the graph unless a large number of local edges is added.

**Related work on spectral rewiring methods.** A different class of approaches consist of rewiring the graph based on a global *spectral* quantity rather than using spatial distance. Two prototypical measures that have been explored in this regard are spectral gap (Karhadkar et al., 2022) and effective

Table 1: Properties of different types of rewirings.

| Property | Spatial | Spectral | **LASER** |
|---|---|---|---|
| Reduce over-squashing | ✓ | ✓ | ✓ |
| Preserve locality | ✓ | ✗ | ✓ |
| Preserve sparsity | ✗ | ✓ | ✓ |

resistance (Arnaiz-Rodríguez et al., 2022; Banerjee et al., 2022; Black et al., 2023). It has recently been shown that a node $v$ is mostly insensitive to information contained at nodes that have high effective resistance (Black et al., 2023; Di Giovanni et al., 2023); accordingly, spectral rewiring approaches alleviate over-squashing by reducing the effective resistance. Moreover, they achieve that adding only a few edges by optimally increasing the chosen measure of connectivity, hence maintaining the sparsity level of the input graph. However, the edges that are added in the graph typically end up connecting very distant nodes (since the distance between two nodes is at least as large as their effective resistance), hence *rapidly* diminishing the role of *locality* provided by distance on the original graph.

**An ideal rewiring approach.** Given a graph $\mathsf{G}$, an ideal rewiring map $\mathcal{R}$ should satisfy the following desiderata: (i) **Reduce over-squashing:** $\mathcal{R}$ increases the overall connectivity of $\mathsf{G}$ — according to some topological measure — in order to alleviate over-squashing; (ii) **Preserve locality:** $\mathcal{R}$ preserves some inductive bias afforded by $\mathsf{G}$, e.g., nodes that are "distant" should be kept separate from nodes that are closer in the GNN architecture; (iii) **Preserve sparsity:** $\mathcal{R}$ approximately preserves the sparsity of $\mathsf{G}$, ideally adding a number of edges linear in the number of nodes. While condition (i) represents the main rationale for rewiring the input graph, criteria (ii) and (iii) guarantee that the rewiring is efficient and do not allow the role played by the structural information in the input graph to degrade too much. As discussed above and summarized in Table 1, spatial methods typically satisfy only (i) and (ii), but not (iii), while spectral-methods meet (i) and (iii) but fail (ii).

**Main idea.** Our main contribution is a novel paradigm for graph rewiring that satisfies criteria (i)–(iii), leveraging a key principle: instead of considering a *single* rewired graph $\mathcal{R}(\mathsf{G})$, we use a *sequence* of rewired graphs $\{\mathcal{R}_\ell(\mathsf{G})\}_\ell$ such that for smaller $\ell$, the new edges added in $\mathcal{R}_\ell(\mathsf{G})$ are more 'local' (with respect to the input graph $\mathsf{G}$) and sampled based on optimizing a connectivity measure.

## 4    A GENERAL PARADIGM: DYNAMIC REWIRING WITH LOCAL CONSTRAINTS

In this Section, we discuss a general graph-rewiring paradigm that can enhance any MPNN and satisfies the criteria (i)–(iii) described above. Given a graph $\mathsf{G}$, consider a trajectory of rewiring operations $\mathcal{R}_\ell$, starting at $\mathsf{G}_0 = \mathsf{G}$, of the form:

$$\mathsf{G} = \mathsf{G}_0 \xrightarrow{\mathcal{R}_1} \mathsf{G}_1 \xrightarrow{\mathcal{R}_2} \cdots \xrightarrow{\mathcal{R}_L} \mathsf{G}_L. \tag{3}$$

Since we think of $\mathsf{G}_\ell$ as the input graph evolved along a dynamical process for $\ell$ iterations, we refer to $\mathsf{G}_\ell$ as the $\ell$-*snapshot*. For the sake of simplicity, we assume $\mathcal{R}_\ell = \mathcal{R}$, though it is straightforward to extend the discussion below to the more general case. In order to account for the multiple snapshots, we modify the layer form in equation 2 as follows:

$$\mathbf{x}_v^{(t)} = \mathsf{up}^{(t)}\left(\mathbf{x}_v^{(t-1)}, \left(\mathsf{a}_{\mathsf{G}_\ell}^{(t)}(\{\!\{\mathbf{x}_u^{(t-1)} : (v, u) \in \mathsf{E}_\ell\}\!\})\right)_{0 \le \ell \le L}\right). \tag{4}$$

Below we describe a rewiring paradigm based on an arbitrary ***connectivity measure*** $\mu : \mathsf{V} \times \mathsf{V} \to \mathbb{R}$ and ***locality measure*** $\nu : \mathsf{V} \times \mathsf{V} \to \mathbb{R}$. The measure $\mu$ can be any topological quantity that captures how easily different pairs of nodes can communicate in a graph, while the measure $\nu$ is any quantity that penalizes interactions among nodes that are 'distant' according to some metric on the input graph. In a nutshell, our choice of $\mathcal{R}$ samples edges to add according to the constraint $\nu$, prioritizing those that maximally benefit the measure $\mu$. By keeping this generality, we provide a universal approach to do graph-rewiring that can be of interest independently of the specific choices of $\mu$ and $\nu$.

**Improving connectivity while preserving locality.** The first property we demand of the rewiring sequence is that for all nodes $v, u$, we have $\mu_{\mathsf{G}_{\ell+1}}(v, u) \geq \mu_{\mathsf{G}_\ell}(v, u)$ and that for *some* nodes, the inequality is *strict*. If we connect all pairs of nodes with low $\mu$-value, however, we might end up adding non-local edges across distant nodes, hence quickly corrupting the locality of $\mathsf{G}$. To avoid this, we *constrain* each rewiring by requiring the measure $\nu$ to take values in a certain range $\mathcal{I}_\ell \subset [0, \infty)$: an edge $(v, u)$ appears in the $\ell$-snapshot (for $1 \leq \ell \leq L$) according to the following rule:

$$(v, u) \in \mathsf{E}_\ell \text{ if } \left( \mu_{\mathsf{G}_0}(v, u) < \epsilon \text{ and } \nu_{\mathsf{G}_0}(v, u) \in \mathcal{I}_\ell \right) \text{ or } (v, u) \in \mathsf{E}_{\ell-1}. \tag{5}$$

To make the rewiring more efficient, the connectivity and locality measures are computed *once* over the input graph $\mathsf{G}_0$. Since the edges to be added connect nodes with low $\mu$-values, the rewiring makes the graphs $\mathsf{G}_\ell$ friendlier to message-passing as $\ell$ grows. Moreover, by taking increasing ranges of values for the intervals $\mathcal{I}_\ell$, we make sure that new edges connect distant nodes, as specified by $\nu$, only at later snapshots. Sequential rewiring allows us to interpolate between the given graph and one with better connectivity, creating intermediate snapshots that *progressively* add non-local edges. By accounting for all the snapshots $\mathsf{G}_\ell$ in equation 2, the GNN can access both the input graph, and more connected ones, at a much *finer level* than 'instantaneous' rewirings, defined next.

**Instantaneous vs sequential rewiring.** As discussed in Section 3, existing rewiring techniques — particularly those of the spectral type — often consider the simpler trajectory $\mathsf{G}_0 \hookrightarrow \mathcal{R}(\mathsf{G}_0) := \mathsf{G}_1$ ("instantaneous rewiring"). The main drawback of this approach is that in order to improve the connectivity in a *single snapshot*, the rewiring map $\mathcal{R}$ is bound to either violate the locality constraint $\nu$, by adding edges between very distant nodes, or compromise the graph-sparsity by adding a large volume of (local) edges. In fact, if that were not the case, we would still be severely affected by over-squashing. Conversely, sequential rewiring allows a *smoother* evolution from the input graph $\mathsf{G}_0$ to a configuration $\mathsf{G}_L$ which is more robust to over-squashing, so that we can more eeasly preserve the inductive bias afforded by the topology via local constraints under equation 5.

**An equivalent perspective: multi-relational GNNs.** In Karhadkar et al. (2022) the notion of relational rewiring was introduced for spectral methods. We expand upon this idea, by noticing that the general, sequential rewiring paradigm described above can be instantiated as a family of multi-relational GNNs (Battaglia et al., 2018; Barcelo et al., 2022). To this aim, consider a slightly more specific instance of equation 4, which extends common MPNN frameworks:

$$\mathbf{x}_v^{(t)} = \mathsf{up}^{(t)} \left( \mathbf{x}_v^{(t-1)}, \sum_{\ell=0}^L \sum_{\substack{u: \\ (v,u) \in \mathsf{E}_\ell}} \psi_\ell^{(t)}(\mathbf{x}_v^{(t-1)}, \mathbf{x}_u^{(t-1)}) \right), \tag{6}$$

where $\psi_\ell^{(t)}$ are learnable message functions depending on both the layer $t$ and the snapshot $\ell$. It suffices now to note that each edge set $\mathsf{E}_\ell$, originated from the rewiring sequence, can be given its own *relation*, so that equation 6 is indeed equivalent to the multi-relation GNN framework of Battaglia et al. (2018). In fact, since we consider rewiring operations that only add edges to improve the connectivity, we can rearrange the terms and rename the update and message-function maps, so that we aggregate over existing edges once, and separately over the newly added edges i.e. the set $\mathsf{E}_\ell \setminus \mathsf{E}_{\ell-1}$. Namely, we can rewrite equation 6 as

$$\mathbf{x}_v^{(t)} = \mathsf{up}^{(t)} \left( \mathbf{x}_v^{(t-1)}, \sum_{u: (v,u) \in \mathsf{E}} \psi_0^{(t)}(\mathbf{x}_v^{(t-1)}, \mathbf{x}_u^{(t-1)}) + \sum_{\ell=1}^L \sum_{\substack{u: \\ (v,u) \in \mathsf{E}_\ell \setminus \mathsf{E}_{\ell-1}}} \psi_\ell^{(t)}(\mathbf{x}_v^{(t-1)}, \mathbf{x}_u^{(t-1)}) \right). \tag{7}$$

Accordingly, we see how our choice of sequential rewiring can be interpreted as an *extension* of relational rewiring in Karhadkar et al. (2022), where $L = 1$. Differently from Karhadkar et al. (2022), the multiple relations $\ell \geq 1$ allow us to add connections over the graph among increasingly less local nodes, meaning that the edge-type $\ell$ is now associated to a notion of locality specified by the choice of the constraint $\nu(v, u) \in \mathcal{I}_\ell$. We finally observe that the connection between graph-rewiring and relational GNNs is not surprising once we think of the sequence of rewiring in equation 3 as snapshots of a *temporal dynamics* over the graph connectivity. Differently from the setting of *temporal GNNs* (Rossi et al., 2020) though, here the evolution of the connectivity over time is guided by our rewiring procedure rather than by an intrinsic law on the data. In fact, Gao & Ribeiro (2022) studied the equivalence between temporal GNNs and static multi-relational GNNs, which further motivate the analogy discussed above.

## 5  LOCALITY-AWARE SEQUENTIAL REWIRING: THE LASER FRAMEWORK

We consider an instance of the outlined sequential rewiring paradigm, giving rise to the **LASER** framework used in our experiments. We show that **LASER** (i) mitigates over-squashing, (ii) preserves the inductive bias provided by the shortest-walk distance on G better than spectral approaches, while (iii) being sparser than spatial-rewiring methods.

**The choice of locality.**  We choose $\nu$ to be the *shortest-walk distance* $d_G$. In particular, if in equation 5 we choose intervals $\mathcal{I}_\ell = \delta_{\ell+1}$, then at the $\ell$-snapshot $G_\ell$ we only add edges among nodes at distance *exactly* $\ell + 1$. Our constraints prevent distant nodes from interacting at earlier snapshots and allows the GNN to learn message functions $\psi_\ell$ in equation 7 for each hop level $\ell$. If we choose $E_\ell \setminus E_{\ell-1}$ to be the set of *all* edges connecting nodes whose distance is *exactly* $\ell + 1$, then equation 7 is equivalent to the $L$-hop MPNN class studied in Feng et al. (2022). This way though, we generally lose the sparsity of G and increase the risk of over-smoothing. Accordingly, we propose to only add edges that satisfy the locality constraint and have connectivity measure 'small' so that their addition is optimal for reducing over-squashing.

**The choice of the connectivity measure $\mu$.**  Although edge curvature or effective resistance R are related to over-squashing (Topping et al., 2022; Black et al., 2023; Di Giovanni et al., 2023), computing these metrics incur high complexity – $O(|E|d_{\max}^2)$ for the curvature and $O(n^3)$ for R. Because of that, we propose a more efficient connectivity measure:

$$\mu_k(v, u) := (\tilde{\mathbf{A}}^k)_{vu}, \quad \tilde{\mathbf{A}} := \mathbf{A} + \mathbf{I}. \tag{8}$$

Because of the self-loops, the entry $(\tilde{\mathbf{A}}^k)_{vu}$ equals the number of walks from $v$ to $u$ of length *at most* $k$. Once we fix a value $k$, if $\mu_k(v, u)$ is large, then the two nodes $v, u$ have multiple alternative routes to exchange information (up to scale $k$) and would usually have small effective resistance. In particular, according to Di Giovanni et al. (2023, Theorem 4.1), we know that the number of walks among two nodes is a *proxy* for how sensitive a pair of nodes is to over-squashing.

**LASER focus.**  We can now describe our framework. Given a node $v$ and a snapshot $G_\ell$, we consider the set of nodes at distance exactly $\ell + 1$ from $v$, which we denote by $\mathcal{N}_{\ell+1}(v)$. We introduce a global parameter $\rho \in (0, 1]$ and add edges (with relation type $\ell$ as per equation 7) among $v$ and the fraction $\rho$ of nodes in $\mathcal{N}_{\ell+1}(v)$ with the lowest connectivity score – if this fraction is smaller than one, then we round it to one. This way, we end up adding only a percentage $\rho$ of the edges that a normal multi-hop GNNs would have, but we do so by *prioritizing those edges that improve the connectivity measure the most*. To simplify the notations, we let $\mathcal{N}_{\ell+1}^\rho(v) \subset \mathcal{N}_{\ell+1}(v)$, be the $\rho$-fraction of nodes at distance $\ell + 1$ from $v$, where $\mu_k$ in equation 8 takes on the lowest values. We express the layer-update of **LASER** as

$$\mathbf{x}_v^{(t)} = \mathsf{up}^{(t)}\Big(\mathbf{x}_v^{(t-1)}, \sum_{u:\,(v,u)\in E} \psi_0^{(t)}(\mathbf{x}_v^{(t-1)}, \mathbf{x}_u^{(t-1)}) + \sum_{\ell=1}^{L} \sum_{u\in\mathcal{N}_{\ell+1}^\rho(v)} \psi_\ell^{(t)}(\mathbf{x}_v^{(t-1)}, \mathbf{x}_u^{(t-1)})\Big). \tag{9}$$

We note that when $\rho = 0$, equation (9) reduces to a standard MPNN on the input graph, while for $\rho = 1$ we recover multi-relational $L$-hop MPNNs (Feng et al., 2022). Although the framework encompasses different choices of the message-functions $\psi_\ell$, in the following we focus on the **LASER**-GCN variant, whose update equation is reported in Appendix (Section A).

We now show that the **LASER** framework satisfies the criteria (i)–(iii) introduced in Section 3. Let $\mathbf{J}^{(r)}(v, u) := \partial\mathbf{x}_v^{(r)}/\partial\mathbf{x}_u^{(0)}$ be the Jacobian of features after $r$ layers of GCN on G, and similarly we let $\tilde{\mathbf{J}}^{(r)}(v, u)$ be the Jacobian of features after $r$ layers of **LASER**-GCN in equation 10. In the following, we take the expectation with respect to the Bernoulli variable ReLU′ which is assumed to have probability of success $\rho$ for all paths in the computational graph as in Xu et al. (2018); Di Giovanni et al. (2023). We recall that given $i \in V$ and $1 \leq \ell \leq L$, $d_{i,\ell}$ enters equation 10.

**Proposition 5.1.** *Let $v, u \in V$ with $d_G(v, u) = r$, and assume that there exists a single path of length $r$ connecting $v$ and $u$. Assume that **LASER** adds an edge between $v$ and some node $j$ belonging to the path of length $r$ connecting $v$ to $u$, with $d_G(v, j) = \ell < r$. Then for all $m \leq r$, we have*

$$\|\mathbb{E}[\tilde{\mathbf{J}}^{(r-\ell+1)}(v, u)]\| \geq \frac{(d_{\min})^\ell}{\sqrt{d_{v,\ell-1}d_{j,\ell-1}}} \|\mathbb{E}[\mathbf{J}^{(m)}(v, u)]\|.$$

The result is not surprising and shows that in general, the **LASER**-rewiring can improve the Jacobian sensitivity significantly and hence alleviates over-squashing, satisfying desideratum (i). Next, we validate that the effects of the local constraints when compared to unconstrained, global spectral methods. Below, we let $\mathcal{D}_{\mathsf{G}}$ be the *matrix of pairwise distances* associated with the graph $\mathsf{G}$, i.e. $(\mathcal{D}_{\mathsf{G}})_{vu} = d_{\mathsf{G}}(v, u)$. We propose to investigate $\|\mathcal{D}_{\mathsf{G}} - \mathcal{D}_{\mathcal{R}(\mathsf{G})}\|_F$, where $\|\cdot\|_F$ is the Frobenius norm and $\mathcal{R}(\mathsf{G})$ is either a baseline spectral rewiring, or our **LASER**-framework. We treat this quantity as a proxy for how well a rewiring framework is able to preserve the inductive bias given by the input graph. In fact, for many graphs (including molecular-type with small average degree), spectral rewirings incur a larger Frobenius deviation even if they add fewer edges, since these edges typically connect very distant nodes in the graph. To this aim, we show a setting where **LASER** preserves more of the locality inductive bias than spectral-based methods provided we choose the factor $\rho$ small enough. Below, we focus on a case that, according to Di Giovanni et al. (2023); Black et al. (2023), we know to be a worst-case scenario for over-squashing considering that the commute time scales cubically in the number of nodes. Put differently, the graph below represents a prototypical case of 'bottleneck' encountered when information has to travel from the end of the chain to the clique.

**Proposition 5.2.** *Let $\mathsf{G}$ be a 'lollipop' graph composed of a chain of length $L$ attached to a clique of size $n$ sufficiently large. Consider a spectral rewiring $\mathcal{R}$ which adds an edge between nodes with the highest effective resistance. We can choose the factor $\rho \in (0, 1)$ as a function of $L$ so that **LASER** with a single snapshot, on average, adds a number of edges that guarantees:*

$$\|\mathcal{D}_{\mathsf{G}} - \mathcal{D}_{\mathcal{R}(\mathsf{G})}\|_F \geq \|\mathcal{D}_{\mathsf{G}} - \mathcal{D}_{\textbf{LASER}}\|_F.$$

We refer to the Appendix (Section A) for an explicit characterization on how large $n$ needs to be depending on $L$ and the proofs of the statements stated above. Finally, as desired in (iii), we observe that compared to dense multi-hop GNNs, LASER is more efficient since it only adds a fraction $\rho$ of edges for each node $v$ and each orbit-level $\mathcal{N}_{\ell+1}(v)$. In fact, for many sparse graphs (such as molecular ones) the model ends up adding a number of edges proportional to the number of nodes (see Section C.2 in the Appendix for a discussion and ablations).

## 6 EXPERIMENTS

In this section, we validate our claims on a range of tasks and benchmarks. Beyond comparing the performance of **LASER** to existing baselines, we run ablations to address the following *important questions*: (1) Does **LASER** improve the graph's connectivity? (2) Does **LASER** preserve locality information better than spectral rewiring approaches? (3) What is the impact of the fraction $\rho$ of edges sampled? (4) What if we sample edges to be added from $\mathcal{N}_{\ell+1}(v)$ randomly, rather than optimally according to $\mu$ in equation 8? (5) Is **LASER** scalable to large graphs? In the Appendix (Section C), we provide a density comparison between **LASER** and Multi-Hop GNNs, discuss our tie-breaking procedure that guarantees equivariance in expectation and further improves performance, provide an ablation using different underlying MPNNs, and discuss additional motivation for the need for locality. We also provide, in Section D, a more thorough scalability analysis.

**Benchmarks.** We evaluate on the Long Range Graph Benchmark (LRGB) (Dwivedi et al., 2022) and TUDatasets (Morris et al., 2020). In the experiments, we fix the underlying model to GCN, but provide ablations with different popular MPNNs in the Appendix (Section C.3). For spatial curvature-based rewirings, we compare against SDRF (Topping et al., 2022) and BORF (Nguyen et al., 2023). For spectral techniques, we compare against FOSR (Karhadkar et al., 2022), a spectral gap rewiring technique, and GTR (Black et al., 2023), an effective resistance rewiring technique. We also compare to DiffWire (Arnaiz-Rodríguez et al., 2022), a differentiable rewiring technique.

Table 2: Results for the `Peptides-func`, `Peptides-struct`, and `PCQM-Contact` datasets. Performances are Average Precision (AP) (higher is better), Mean Absolute Error (MAE) (lower is better), and Mean Reciprocal Rank (MRR) (higher is better), respectively.

| Rewiring | Peptides-func | Peptides-struct | PCQM-Contact |
|---|---|---|---|
| | Test AP ↑ | Test MAE ↓ | Test MRR ↑ |
| None | 0.5930±0.0023 | 0.3496±0.0013 | 0.3234±0.0006 |
| SDRF | 0.5947±0.0035 | 0.3404±0.0015 | 0.3249±0.0006 |
| GTR | 0.5075±0.0029 | 0.3618±0.0010 | 0.3007±0.0022 |
| FOSR | 0.5947±0.0027 | 0.3078±0.0026 | 0.2783±0.0008 |
| BORF | 0.6012±0.0031 | 0.3374±0.0011 | TIMEOUT |
| **LASER** | **0.6440**±0.0010 | **0.3043**±0.0019 | **0.3275**±0.0011 |

Based on Karhadkar et al. (2022) and the parallelism we draw between rewiring and multi-relational GNNs, for all techniques, we report results tuned over both a 'standard' and relational (Schlichtkrull et al., 2018) model for the baselines, where we assign original and rewired edges distinct relational types. In particular, R-GCN in these cases is then a special instance of equation 2. For additional details on the tasks and hyper-parameters, we refer to the Appendix (Section B).

**LRGB.** We consider the `Peptides` (15 535 graphs) and `PCQM-Contact` (529 434 graphs) datatsets, from the Long Range Graph Benchmark (LRGB). There are two tasks associated with `Peptides`, a peptide function classification task `Peptides-func` and a peptide structure regression task `Peptides-struct`. `PCQM-Contact` is a link-prediction task, in which the goal is to predict pairs of distant nodes that will be adjacent in 3D space. We replicate the experimental settings from Dwivedi et al. (2022), with a 5-layer MPNN for each of the rewirings as the underlying model. We choose the hidden dimension in order to respect the 500k parameter budget. In Table 2, we report the performance on the three tasks. **LASER** convincingly outperforms all baselines on the three tasks, while the other rewiring baselines frequently perform worse than the standard GCN model. On `PCQM-Contact`, the rewiring time for BORF surpasses the 60 hour limit enforced by Dwivedi et al. (2020) on our hardware, so we assign it a TIMEOUT score.

**TUDatasets.** We evaluate **LASER** on the `REDDIT-BINARY`, `IMDB-BINARY`, `MUTAG`, `ENZYMES`, `PROTEINS`, and `COLLAB` tasks from TUDatasets, which were chosen by Karhadkar et al. (2022) under the claim that they require long-range interactions. We evaluate on 25 random splits, fixing the hidden dimension for all models to 64 and the number of layers to 4, as in Karhadkar et al. (2022). We avoid the use of dropout and use Batch Norm (Ioffe & Szegedy, 2015). We refer to the Appendix (Section B.2) for further details on the hyper-parameters and a discussion on some drawbacks of these tasks. Table 3 shows the results on the aforementioned benchmarks. **LASER** most consistently achieves the best classification accuracy, attaining the highest mean rank.

Table 3: Accuracy ± std over 25 random splits for the datasets and rewirings. Colors highlight First, Second, and Third; we report the mean rank achieved on the valid runs. OOM is Out of Memory.

| Rewiring | REDDIT-BINARY | IMDB-BINARY | MUTAG | ENZYMES | PROTEINS | COLLAB | Mean Rank |
|---|---|---|---|---|---|---|---|
| None | 81.000±2.717 | 64.280±1.990 | 74.737±5.955 | 28.733±5.297 | 64.286±2.004 | 68.960±2.284 | 4.83 |
| DiffWire | OOM | 59.000±3.847 | 80.421±9.707 | 28.533±4.475 | 72.714±2.946 | 65.440±2.177 | 4.83 |
| GTR | 85.700±2.786 | 52.560±4.104 | 78.632±6.201 | 26.333±5.821 | 72.303±4.658 | 68.024±2.299 | 4.67 |
| SDRF | 84.420±2.785 | 58.290±3.201 | 74.526±5.355 | 30.567±6.188 | 68.714±4.233 | 70.222±2.571 | 4.50 |
| FOSR | 85.930±2.793 | 60.400±5.855 | 75.895±7.211 | 28.600±5.253 | 71.643±3.428 | 69.848±3.485 | 3.67 |
| BORF | 84.920±2.534 | 60.820±3.877 | 81.684±7.964 | 30.500±6.593 | 68.411±4.122 | OOM | 3.60 |
| **LASER** | 85.458±2.827 | 64.333±3.298 | 82.204±6.728 | 34.333±6.936 | 74.381±3.443 | 70.923±2.538 | 1.37 |

**Ablation studies.** In the following, we choose FOSR as a typical spectral rewiring approach, while taking **LASER** with $\rho = 1$ as an instance of a dense, multi-hop GNN (i.e. classical spatial rewiring). For the purpose of these ablations, we conduct experiments on the `Peptides` dataset. We start by investigating questions (1) and (2), namely, how well **LASER** improves connectivity while respecting locality. To this end, we increment the number of snapshots from 2 to 5 given densities $\rho = 0.1$ and $\rho = 1$ for **LASER** and instead vary the number of edge additions of FOSR spanning the values 10, 20, 50, and 100. To assess the connectivity, we report the mean total effective resistance — which is a good proxy for over-squashing (Black et al., 2023; Di Giovanni et al., 2023) — while for the locality, we evaluate the norm of the difference between the original graph distance matrix and that of the rewired graph $\|\mathcal{D}_\mathsf{G} - \mathcal{D}_{\mathcal{R}(\mathsf{G})}\|_F$ as per Proposition 5.2. Figure 2 shows the results of this ablation. We validate that the sparse **LASER** framework decreases the mean total effective resistance consistently over increasing snapshots as well as other rewiring techniques. Moreover, we find that **LASER** with $\rho = 0.1$ is *better* than dense spatial methods and especially surpasses spectral approaches *at preserving information contained in the distance matrix.*

Next, we investigate question (3), i.e. the impact of the fraction $\rho$ of edges being sampled, by increasing the number of snapshots from 2 to 5 and varying the density $\rho$ ranging 0.1, 0.25, 0.5, and 1, with results reported in Figure 3. The majority of the *performance gains are obtained through a sparse rewiring*, as even with $\rho = 0.1$

Table 4: Comparison between **LASER** and random sampling, with $L = 3$ and $\rho = 0.1$.

| Model | Peptides-func ↑ | Peptides-struct ↓ |
|---|---|---|
| Random | 0.4796±0.0067 | 0.3382±0.0019 |
| **LASER** | **0.6414**±0.0020 | **0.3119**±0.0005 |

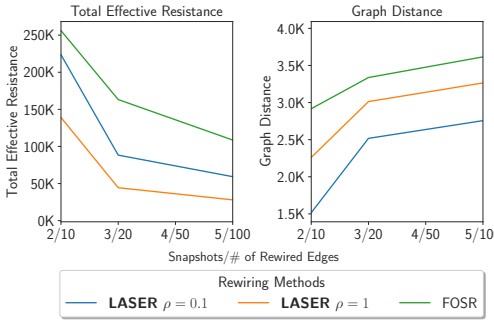 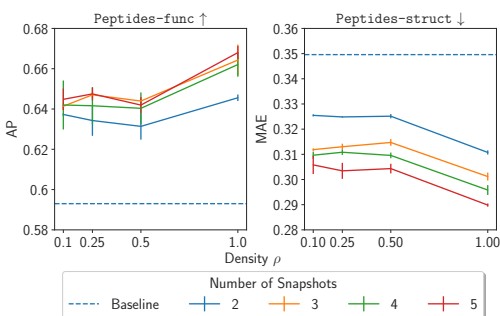

Figure 2: Total effective resistance and graph distance, as measured by the Frobenius norm, when varying the number of snapshots from 2 to 5 and $\rho$ from 0.1 to 1.

Figure 3: Performance on the `Peptides` tasks when varying the number of snapshots from 2 to 5 and $\rho$ from 0.1 to 1. The baseline indicates a standard GCN model.

the performance is greatly increased over the baseline. The additional density in the orbits does seem to help with performance, but this comes at the cost of density.

Finally, we address question (4), by evaluating how sampling edges uniformly over the nodes at distance $\ell + 1$ given a density $\rho$, compares to our choice of prioritizing edges with lowest connectivity score $\mu$ as per equation 8. We report the results in Table 4. We see that **LASER** greatly outperforms the random rewiring, verifying our claim that guiding the rewiring through $\mu$ is a more sound approach.

**Scalability.** The operations required to compute $\mu$ and $\nu$ in **LASER** are designed to be efficiently implemented on modern hardware accelerators, mostly relying on matrix multiplication. Furthermore, the rewiring operation is done once and stored for future runs. The $\rho$ factor can be tuned to calibrate the density of the rewiring, giving further control on the training efficiency. **LASER** does not seem to significantly impact the run-time compared to the standard baseline models and we found through a synthetic benchmarking experiment that our implementation of **LASER** *is able to rewire graphs with 100k nodes and a million edges in 2 hours*. This is in contrast to FOSR and SDRF that failed to finish the computation within 24 hours. We report a large number of benchmarking experiments, alongside a theoretical complexity analysis in the Appendix (Section D).

## 7   CONCLUSION

In this work, we have identified shortcomings of rewiring techniques and argued that a rewiring must: (i) improve connectivity, (ii) respect locality, and (iii) preserve sparsity. Unlike current spectral and spatial rewirings that compromise some of these properties, we have outlined a general rewiring paradigm that meets criteria (i)–(iii) by interpolating between the input graph and a better connected one via locally constrained sequential rewiring. We have then proposed a specific instance of this paradigm — **LASER** — and verified, both theoretically and empirically, that it satisfies (i)-(iii).

**Limitations and Future Work.**   In this paper, we considered a simple instance of the general rewiring paradigm outlined in Section 4, but we believe that an interesting research direction would be to explore alternative choices for both the connectivity and locality measures, ideally incorporating features in a differentiable pipeline similar to Arnaiz-Rodríguez et al. (2022). Furthermore, the identification between graph-rewiring on the one hand, and multi-relational GNNs and temporal-GNNs on the other, could lead to interesting connections between the two settings, both theoretically (e.g., what is the expressive power of a certain rewiring policy?) and practically, where techniques working in one case could be effortlessly transferred to the other. Finally, we highlight that, as is customary in rewiring approaches, it is always hard to pinpoint with certainty the reason for any performance improvement, including whether such an improvement can be truly credited to over-squashing and long-range interactions. We have tried to address this point through multiple ablations studies.

## 8 ACKNOWLEDGEMENTS

FdG, FB, and MB are partially supported by the EPSRC Turing AI World-Leading Research Fellowship No. EP/X040062/1. We would like to thank Google Cloud for kindly providing computational resources for this work.

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

## A ADDITIONAL DETAILS ON THEORY AND FRAMEWORK

In this Section, we report additional considerations about our framework and provide proofs for the theoretical results stated in Section 5.

First, we report the variant of GCN-**LASER** to make things more concrete:

$$\mathbf{x}_v^{(t)} = \mathsf{ReLU}\Big( \sum_{u \in \mathcal{N}_1(v) \cup \{v\}} \frac{1}{\sqrt{d_v d_u}} \mathbf{W}_0^{(t)} \mathbf{x}_u^{(t-1)} + \sum_{\ell=1}^{L} \sum_{u \in \mathcal{N}_{\ell+1}^\rho(v)} \frac{1}{\sqrt{d_{v,\ell} d_{u,\ell}}} \mathbf{W}_\ell^{(t)} \mathbf{x}_u^{(t-1)} \Big), \quad (10)$$

where $\mathbf{W}_\ell^{(t)}$, for $0 \leq \ell \leq L$, are learnable weight matrices, while $d_{i,\ell}$ is equal to the degree induced by the distance matrix associated with the value $\ell$, for each node $i \in \mathsf{V}$. We base most of our evaluation on the model in equation 10.

**Proposition A.1.** *Let $v, u \in \mathsf{V}$ with $d_\mathsf{G}(v, u) = r$, and assume that there exists a single path of length $r$ connecting $v$ and $u$. Assume that **LASER** adds an edge between $v$ and some node $j$ belonging to the path of length $r$ connecting $v$ to $u$, with $d_\mathsf{G}(v, j) = \ell < r$. Finally, assume for simplicity that all products of weight matrices have unit norm. Then for all $m \leq r$, we have*

$$\|\mathbb{E}[\tilde{\mathbf{J}}^{(r-\ell+1)}(v, u)]\| \geq \frac{(d_{\min})^\ell}{\sqrt{d_{v,\ell-1} d_{j,\ell-1}}} \|\mathbb{E}[\mathbf{J}^{(m)}(v, u)]\|$$

*Proof.* We first note that extending the result to arbitrary weight matrices is trivial, since one would just obtain an extra factor in the lower bound of the form $\omega^{r-\ell+1}/(\omega')^m$ depending on the spectral bounds (i.e. singular values bounds) of the weight matrices entering **LASER** -GCN and GCN, respectively.

Following the assumptions, we can argue precisely as in Xu et al. (2018) and Di Giovanni et al. (2023, Section 5) and write the Jacobian of node features after $m$ layers of GCN (Kipf & Welling, 2017) as

$$\mathbb{E}[\mathbf{J}^{(m)}(v, u)] = \rho \prod_{k=1}^{m} \mathbf{W}^{(k)} (\hat{\mathbf{A}}^m)_{vu},$$

where $\hat{\mathbf{A}} = \mathbf{D}^{-1/2} \mathbf{A} \mathbf{D}^{-1/2}$ is the symmetrically normalized adjacency matrix. In particular, we can then estimate the norm of the Jacobian matrix simply by:

$$\|\mathbb{E}[\mathbf{J}^{(m)}(v, u)]\| = \rho \|\prod_{k=1}^{m} \mathbf{W}^{(k)}\| (\hat{\mathbf{A}}^m)_{vu} = \rho(\hat{\mathbf{A}}^m)_{vu},$$

where we have used our simplifying assumption on the norm of the weights. In particular, since $d_\mathsf{G}(v, u) = r$, if $m \leq r$ then the term above vanishes which satisfies the lower bound in the claim. Let us then consider the case $m = r$. We can write the unique path of length $r$ connecting $v$ and $u$ as a tuple $(v, u_1, \ldots, u_{r-1}, u)$, so that

$$\|\mathbb{E}[\mathbf{J}^{(m)}(v, u)]\| = \rho \frac{1}{\sqrt{d_v d_u}} \prod_{s=1}^{r-1} \frac{1}{d_s}.$$

Similarly, given our **LASER** -GCN framework in equation 10 and the assumptions, we can bound the norm of the expected Jacobian as

$$\|\mathbb{E}[\tilde{\mathbf{J}}^{(r-\ell+1)}(v, u)]\| = \rho \| \prod_{k=2}^{r-\ell+1} \mathbf{W}_0^{(k)} \mathbf{W}_{\ell-1}^{(1)}\| (\hat{\mathbf{A}}_{\ell-1})_{vj} (\hat{\mathbf{A}}^{m-\ell})_{ju} = \rho(\hat{\mathbf{A}}_{\ell-1})_{vj} (\hat{\mathbf{A}}^{m-\ell})_{ju},$$

where $(\hat{\mathbf{A}}_\ell)_{ij} = 1/\sqrt{d_{i,\ell} d_{j,\ell}}$ as defined in equation 10 if $d_\mathsf{G}(i, j) = \ell + 1$ and zero otherwise. If we now take the ratio of the two expected values, we can bound them from below as

$$\frac{\|\mathbb{E}[\tilde{\mathbf{J}}^{(r-\ell+1)}(v, u)]\|}{\|\mathbb{E}[\mathbf{J}^{(r)}(v, u)]\|} = \frac{(\hat{\mathbf{A}}_{\ell-1})_{vj} (\hat{\mathbf{A}}^{m-\ell})_{ju}}{\frac{1}{\sqrt{d_v d_u}} \prod_{s=1}^{r-1} \frac{1}{d_s}} \geq \frac{(\hat{\mathbf{A}}_{\ell-1})_{vj}}{\frac{1}{\sqrt{d_v d_u}} \prod_{s=1}^{\ell-1} \frac{1}{d_s}} \geq \frac{(d_{\min})^\ell}{\sqrt{d_{v,\ell-1} d_{j,\ell-1}}},$$

where we have used that by assumption there must exist only one path of length $m - \ell$ from $j$ to $u$, which has same degrees $\{d_s\}$. □

**Proposition A.2.** *Let* G *be a 'lollipop' graph composed of a chain of length $L$ attached to a clique of size $n$ sufficiently large. Consider a spectral rewiring $\mathcal{R}$ which adds an edge between nodes with the highest effective resistance. We can choose the factor $\rho \in (0,1)$ as a function of $L$ so that **LASER** with a single snapshot, on average, adds a number of edges that guarantees:*

$$\|\mathcal{D}_G - \mathcal{D}_{\mathcal{R}(G)}\|_F \geq \|\mathcal{D}_G - \mathcal{D}_{\textbf{LASER}}\|_F.$$

*Proof.* Let us denote the end node of the chain by $v$, while $z$ is the node belonging to both the chain and the clique, and $u$ be any node in the interior of the clique. It is known that the commute time between $v$ and $u$ scales cubically in the total number of nodes (Chandra et al., 1996), so an algorithm aimed at minimizing the effective resistance will add an edge between $v$ and a point in the interior of the clique — which we rename $u$ without loss of generality. Accordingly, the distance between $v$ and any point in the interior has changed by at least $(L+1) - 2$. Besides, the distance between $v$ and $z$ has changed by $L - 2$. We can then derive the lower bound:

$$\|\mathcal{D}_G - \mathcal{D}_{\mathcal{R}(G)}\|_F \geq \sqrt{(n-1)(L-1)^2 + (L-2)^2}.$$

Let us now consider the case of **LASER** with a single snapshot. We want to choose $\rho$ sufficiently small so to avoid adding too many edges. In order to do that, let us focus on the chain. Any node in the chain with the exception of the one before $z$, which we call $z'$ (which has the whole clique in its 2-hop), has a 2-hop neighbourhood of size at most 2. Accordingly, given a number of edges $k$ we wish to be adding, if we choose

$$\rho = \frac{k}{2L},$$

it means that our algorithm, on average, will only add $k$ edges over the chain. To avoid vacuous cases, consider $k \geq 2$. Accordingly, the pairwise distance between any couple of nodes along the chain is changed by at most $k$. For what concerns the clique instead, let us take the worst-case scenario where $z'$ is connected to any node in the clique. Then, the distance between any node in the clique and any node in the chain has changed by $(k+1)$. If we put all together, we have shown that

$$\|\mathcal{D}_G - \mathcal{D}_{\textbf{LASER}}\|_F \leq \sqrt{L^2 k^2 + nL(k+1)^2}.$$

Therefore, the bound we have claimed holds if and only if

$$L^2 k^2 + nL(k+1)^2 \leq (n-1)(L-1)^2 + (L-2)^2. \tag{11}$$

One can now manipulate the inequality and find that the bound is satisfied as long as

$$n \geq \frac{k^2 L^2 + 2L - 3}{L^2(1-k^2) - 3L - 2kL + 1}$$

and note that the denominator is always positive if $k^2 < L/4$ and $L \geq 8$. Accordingly, we conclude that if

$$\rho \leq \frac{\sqrt{L/4}}{L} = \frac{1}{2\sqrt{L}}$$

and $L \geq 8$ our claim holds. In particular, **LASER** will have added $k \in (0, \sqrt{L}/2)$ edges in total. $\qquad\square$

## B EXPERIMENTAL DETAILS

**Reproducibility statement.** We release our code on the following URL https://github.com/Fedzbar/laser-release under the MIT license. For the additional baselines, we borrowed the implementations provided by the respective authors. We slightly amended the implementation of GTR as it would encounter run-time errors when attempting to invert singular matrices on certain graphs.

**Hyper-parameters.** For the LRGB experiments, we use the same hyper-parameters and configurations provided by Dwivedi et al. (2020), respecting a 500k parameter budget in all the experiments. We lightly manually tune the number of snapshots with values $L \in \{2, 3, 4, 5\}$ and the density with values $\{1/10, 1/4, 1/2\}$ for **LASER**. For FOSR, SDRF, and GTR we search the number of iterations from $\{5, 20, 40\}$, similarly to their respective works. For BORF, as the rewiring is much

more expensive, especially on these 'larger' datasets, we fix the number of iterations to 1. We point out that with the implementation provided by the authors, BORF would exceed the 60 hours limit imposed by Dwivedi et al. (2022) on our hardware for `PCQM-Contact` and for this reason we assigned it a TIMEOUT value.

For the TUDatasets experiments, we use ADAM (Kingma & Ba, 2015) with default settings and use the `ReduceLROnPlateau` scheduler with a patience of 20, a starting learning rate of 0.001, a decay factor of $1/2$, and a minimum learning rate of $1 \times 10^{-5}$. We apply Batch Norm (Ioffe & Szegedy, 2015), use ReLU as an activation function, and fix the hidden dimension to 64. We do not use dropout, avoid using a node encoder and use a weak (linear) decoder to more accurately compare the various rewiring methods. We lightly manually tune the number of snapshots with values $L \in \{2, 3\}$ and the density with values $\{1/10, 1/4, 1/2\}$ for **LASER**. For FOSR, SDRF, and GTR we search the number of iterations from $\{5, 20, 40\}$, similarly to their respective works. For BORF, we sweep over $\{1, 2, 3\}$ iterations. For DiffWire, we search between a normalized or Laplacian derivative and set the number of centers to 5.

For both LRGB and the TUDatasets the additional baseline models are also further tuned using either a relational or non-relational GCN. For instance, in the main text we group the results of FOSR and R-FOSR together for clarity. In general, we found relational models to perform better than the non-relational counter-parts. Such a result is consistent with results reported by other rewiring works.

**Hardware.** Experiments were ran on 2 machines with $4\times$ NVIDIA Tesla T4 (16GB) GPU, 16 core Intel(R) Xeon(R) CPU (2.00GHz), and 40 GB of RAM, hosted on the Google Cloud Platform (GCP). For the `PQCM-Contact` experiments we increased the RAM to 80GB and the CPU cores to 30.

### B.1 DATASETS

**LRGB** We consider the `Peptides-struct`, `Peptides-func`, and `PCQM-Contact` tasks from the Long Range Graph Benchmark (LRGB) (Dwivedi et al., 2022). `Peptides-func` (15 535 graphs) is a graph classification task in which the goal is to predict the peptide function out of 10 classes. The performance is measured in Average Precision (AP). `Peptides-struct` (15 535 graphs) is a graph regression task in which the goal is to predict 3D properties of the peptides with the performance being measured in Mean Absolute Error (MAE). `PCQM-Contact` (529 434 graphs) is a link-prediction task in which the goal is to predict pairs of distant nodes (when considering graph distance) that are instead close in 3D space. In our experiments, we follow closely the experimental evaluation in Dwivedi et al. (2022), we fix the number of layers to 5 and fix the hidden dimension as such to respect the 500k parameter budget. For `Peptides` we use a 70%/15%/15% train/test/split, while for `PCQM-Contact` we use a 90%/5%/5% split. We train for 500 epochs on `Peptides` and for 200 on `PCQM-Contact`. We would like to also point out that in work concurrent to ours, Tönshoff et al. (2023) have shown that there are better hyper-parameter configurations than the ones used by Dwivedi et al. (2022) for the LRGB tasks that significantly improve the performance of certain baselines.

**TUDatasets** We consider the `REDDIT-BINARY` (2 000 graphs), `IMDB-BINARY` (1 000 graphs), `MUTAG` (188 graphs), `ENZYMES` (600 graphs), `PROTEINS` (1 113 graphs), and `COLLAB` (5 000 graphs) tasks from TUDatasets (Morris et al., 2020). These datasets were chosen by Karhadkar et al. (2022), under the claim that they require long-range interactions. We train for 100 epochs over 25 random seeds with a 80%/10%/10% train/val/test split. We fix the number of layers to 4 and the hidden dimension to 64 as in Karhadkar et al. (2022). However, unlike Karhadkar et al. (2022), we apply Batch Norm and set dropout to 0% instead of 50%. We also avoid using multi-layered encoders and decoders, in order to more faithfully compare the performance of the rewiring techniques. The reported performance is accuracy $\pm$ standard deviation $\sigma$. We note that Karhadkar et al. (2022) report as an uncertainty value $\sigma/\sqrt{N}$ with $N$ being the number of folds. As they set $N = 100$, they effectively report mean $\pm\sigma/10$. We instead report simply the standard deviation $\sigma$ as we deemed this to be more commonly used within the community.

### B.2 DISCUSSION ON THE TUDATASETS

In Table 3, we show an evaluation of **LASER** on the `REDDIT-BINARY`, `IMDB-BINARY`, `MUTAG`, `ENZYMES`, `PROTEINS`, and `COLLAB` tasks from the TUDatasets, which were chosen by Karhadkar et al. (2022) under the claim that they require long-range interactions. We evaluate on a 80%/10%/10% train/val/test split on 25 random splits. We fix the hidden dimension for all models to 64 and the number of layers to 4 as in Karhadkar et al. (2022), but set dropout rate to 0% instead of 50% as we deem this more appropriate. The goal of the evaluation is to compare the rewiring techniques directly, while the high dropout may complicate a more direct evaluation. We train for 100 epochs. For these experiments, we fix the underlying MPNN to GCN.

We point out that the datasets chosen in the evaluation of Karhadkar et al. (2022) have characteristics that may be deemed problematic. First, many of the datasets contain few graphs. For instance, `MUTAG` contains 188 graphs, meaning that only 18 graphs are part of the test set. Further, `REDDIT-BINARY`, `IMDB-BINARY`, and `COLLAB` do not have node features and are augmented with constant feature vectors. Consequently, is not immediately clear how much long-range interactions play a role in these tasks, or in fact, how to even define over-squashing on graphs without (meaningful) features. For these reasons, we believe the LRGB tasks to be more indicative of the benefits of graph-rewiring to mitigate over-squashing.

Furthermore, the standard deviation of the reported accuracy is relatively large on some of the benchmarks, especially on the smaller `MUTAG` and `ENZYMES` tasks. While we report directly the standard deviation $\sigma$ in our uncertainty quantification, Karhadkar et al. (2022) instead report $\sigma/\sqrt{N}$ with $N$ being the number of folds. In Karhadkar et al. (2022) they set $N = 100$, meaning that effectively they report the standard deviation divided by a factor of 10. The high uncertainty in these datasets can be also seen in the results from Karhadkar et al. (2022). In particular, an accuracy of $\approx 68.3\%$ on `MUTAG` is reported using a GCN without rewiring, while an accuracy of $\approx 49.9\%$ is reported on `MUTAG` with an R-GCN without rewiring. However, the two models should achieve similar accuracy, as the R-GCN without rewiring is equivalent to a standard GCN model as there are no further edge types in `MUTAG`.

For these reasons, we believe that while the results of **LASER** on the TUDatasets are strong and beat the other baselines, any evaluation done on these tasks should be considered with a degree of caution as the high standard deviation and quality issues of the chosen tasks leave the results possibly less conclusive. Regardless, we retain such an evaluation in our work for completeness of comparison with the benchmarks used by Karhadkar et al. (2022).

## C ADDITIONAL RESULTS

In this Section, we provide additional results and ablations. In Section C.1, we discuss the orbit sampling procedure that makes **LASER** permutation-equivariant in expectation. In Section C.2, we provide a density comparison between **LASER** and multi-hop GNNs ($\rho = 1$). In Section C.3, we show that **LASER** is able to work well over a range of popular MPNNs. Finally, in Section C.4, we give further motivation for the need for locality.

### C.1 PERMUTATION-EQUIVARIANCE OF **LASER** .

When selecting the edges that need to be rewired given the connectivity measure $\mu$, care needs to be given when handling the tie-breaks in order to remain permutation-equivariant. Assume we have to select $k$ nodes from a partially ordered set of size $n > k$, given a reference node $v$. Further assume that nodes from $k' + 1 < k$ to $p > k$ have equivalent connectivity measure, i.e. we are given a sequence of nodes $u_1, \ldots, u_n$ such that:

$$\mu(v, u_1) \leq \cdots \leq \mu(v, u_{k'}) < \mu(v, u_{k'+1}) = \cdots = \mu(v, u_p) \leq \cdots \leq \mu(v, u_n).$$

We start by selecting the first $k'$ nodes $u_1, \ldots, u_{k'}$ as they are the ones with lowest connectivity measure. Next, we have to select the remaining $k - k'$ nodes from $u_{k'+1}, \ldots, u_p$. If we naively select the nodes $u_{k'+1}, \ldots, u_k$, we would encounter permutation-equivariance issues as we would be relying on an arbitrary ordering of the nodes. Instead, in **LASER** we sample uniformly the remaining

Table 5: Performance on the `Peptides` tasks given a **LASER** implementation that is not permutation-equivariant (choosing top $k$ based on $\mu$ and tie-breaks are chosen based on node id) and one that is equivariant (through sampling uniformly from the tie-breaks). Bold denotes best.

| Rewiring | Peptides-func ↑ | Peptides-struct ↓ |
|---|---|---|
| **LASER** - Not Equivariant | 0.6385±0.0048 | 0.3162±0.0032 |
| **LASER** | **0.6447**±0.0033 | **0.3151**±0.0006 |

Table 6: Mean added edges per graph given a **LASER** rewiring with a density of $\rho = 0.1$ and $\rho = 1$ on the `Peptides` dataset.

| Density | 2 Snapshots | 3 Snapshots | 4 Snapshots | 5 Snapshots |
|---|---|---|---|---|
| **LASER** $\rho = 0.1$ | 148.9 | 296.5 | 442.4 | 587.2 |
| **LASER** $\rho = 1$ | 205.8 | 434.7 | 691.6 | 986.1 |

$k - k'$ from the nodes $k' + 1, \ldots, p$ that have equivalent connectivity measure, assuring permutation-equivariance in expectation. Table 5 shows that the permutation-equivariant implementation performs better than the naive implementation of simply selecting the first $k$ nodes sorted with respect to $\mu$. In practice this shuffling can be implemented efficiently by sampling a random vector $\mathbf{x} \sim \mathcal{N}(\mathbf{0}, \sigma\mathbf{I})$, with $\sigma$ very small and then adding this to the vector of connectivity measures, i.e. $\hat{\boldsymbol{\mu}} = \boldsymbol{\mu} + \mathbf{x}$. When $\sigma$ is small enough, this has the effect of uniformly breaking the tie-breaks, without breaking the absolute order of the connectivity measures.

## C.2 DENSITY ABLATION.

Table 6 reports the mean added edges per graph (counting undirected edges only once) given $\rho = 0.1$ and $\rho = 1$ on the `Peptides` graphs. We note that regardless of $\rho$, we always add a minimum of 1 edge per node orbit. Given that the (molecular) graphs in peptides are very sparse graphs with an average of 151 nodes and 307 edges, the table highlights a *worst-case scenario* for **LASER** as the orbits grow relatively slowly. Having said this, the number of added edges grows at a significantly lower rate given $\rho = 0.1$. This showcases the use of $\rho$ to control the density of the graphs. The minimum number of node rewirings added being set to 1 is a design choice and this can also be further tuned to control the density, if desired.

## C.3 PERFORMANCE WITH DIFFERENT UNDERLYING MPNNS

We run experiments to evaluate the performance of **LASER** operating over different MPNNs. We evaluate on popular MPNN models: Graph Convolution Networks (GCNs) (Kipf & Welling, 2017), Graph Isomorphism Networks (GINs) (Xu et al., 2019), GAT Veličković et al. (2018), and GraphSAGE (SAGE) Hamilton et al. (2017). Table 7 shows the results on `Peptides-func` and `Peptides-struct` obtained by varying the underlying MPNN. We see that **LASER** improves the baseline MPNN performance consistently reaching best or near best performance. This is not the case for FOSR and SDRF that often end up harming the performance of the baseline MPNN, even when using a relational model. This experiment provides evidence supporting the fact **LASER** is able to function well regardless of the underlying convolution being considered.

## C.4 MOTIVATING LOCALITY

In this section, we motivate the desire for a rewiring technique to respect the locality of the graph. Preserving locality is a natural inductive bias to have whenever we assume that the graph-structure associated with the data is aligned with the downstream task. For instance, molecular systems observe long-range interactions that decay with the distance, in the form of Coulomb electrostatic forces. This behaviour also naturally appears in social networks, transaction networks, or more generally in

Table 7: Results for `Peptides-func` and `Peptides-struct`. Performances are Average Precision (AP) (higher is better) and Mean Absolute Error (MAE) (lower is better), respectively. Results in **bold** denote the best result for each MPNN.

| Model | Peptides-func | Peptides-struct |
|---|---|---|
| | Test AP ↑ | Test MAE ↓ |
| GCN | 0.5930±0.0023 | 0.3496±0.0013 |
| GCN-FOSR-R | 0.4629±0.0071 | 0.3078±0.0026 |
| GCN-SDRF-R | 0.5851±0.0033 | 0.3404±0.0015 |
| GCN-**LASER** | **0.6440**±0.0010 | **0.3043**±0.0019 |
| GIN | 0.5799±0.0006 | 0.3493±0.0007 |
| GIN-FOSR-R | 0.4864±0.0054 | **0.2966**±0.0024 |
| GIN-SDRF-R | 0.6131±0.0084 | 0.3394±0.0012 |
| GIN-**LASER** | **0.6489**±0.0074 | 0.3078±0.0026 |
| GAT | 0.5800±0.0061 | 0.3506±0.0011 |
| GAT-FOSR-R | 0.4515±0.0044 | 0.3074±0.0029 |
| GAT-SDRF-R | 0.5756±0.0037 | 0.3422±0.0008 |
| GAT-**LASER** | **0.6271**±0.0052 | **0.2971**±0.0037 |
| SAGE | 0.5971±0.0041 | 0.3480±0.0007 |
| SAGE-FOSR-R | 0.4678±0.0068 | **0.2986**±0.0013 |
| SAGE-SDRF-R | 0.5892±0.0040 | 0.3408±0.0011 |
| SAGE-**LASER** | **0.6464**±0.0032 | 0.3004±0.0032 |

physical systems, in which interactions that are nearby are more likely to be important for a given task. Accordingly, given a budget of edges to be added, it is sensible to prioritise adding connections between nodes that are closer.

Through a spectral rewiring, one is able to efficiently improve the information flow, but this often greatly modifies the information given by the topology of the graph, as shown in Figure 2. This may be beneficial in tasks in which mixing all information quickly is important, but GNNs usually do not operate under such conditions. In fact, in tasks in which locality 'should' be preserved, spectral rewirings tend to perform poorly, as shown in Table 2. Furthermore, it is unclear to what extent spectral graph-rewirings are able to deal with generalization to varying graph sizes. For instance, a spectral rewiring on a very large peptide chain will connect the most distant parts of the peptide, drastically reducing spatial quantities such as diameter. Instead, on a small molecule this change in topology would be comparably much more tame. On the other hand, local rewiring techniques naturally generalize to different graph sizes as the mechanism via which they alter the topology is much more consistent, being a local procedure.

Figure 4 further motivates the need for locality in **LASER** and supports the claim that locality is the main source of improvement, rather than the more expressive architecture capable of handling a sequence of snapshots. In the experiment, we compare **LASER** to a version of FOSR in which each snapshot contains 10 rewirings. We observe that for **LASER** a larger quantity of snapshots seems to benefit the performance. Instead, with FOSR there is a slight degrade in performance. Regardless, even with a growing number of snapshots, FOSR is not able to compete with **LASER** . This supports the claim that even within our snapshot framework, it is important for the snapshots to remain local, rather than acting globally as done in FOSR.

## D  SCALABILITY AND IMPLEMENTATION

In this section, we provide theoretical and practical insights on the scalability of **LASER** to large graphs. We start by providing a theoretical complexity analysis, followed by benchmarking pre-processing and training times for the various rewiring techniques under comparison on our hardware. We find that the **LASER** rewiring procedure is able to scale to synthetically generated graphs with 100k nodes and a million edges. We further find that the densification due to the rewiring does not

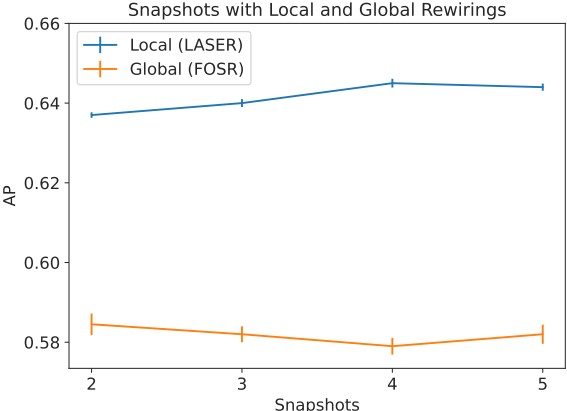

Figure 4: Average Precision (AP) on `Peptides-func` obtained by varying the number of snapshots for **LASER** and FOSR, respecting the 500k parameter budget. For **LASER** , we fix $\rho = 0.1$. For FOSR, each snapshot contains 10 iterations.

significantly impact the total training and inference time when compared to the underlying MPNN without rewirings on real-world tasks.

### D.1 COMPLEXITY ANALYSIS

The computations in **LASER** during the pre-processing time are achieved in two steps: (1) the connectivity measure $\mu$ and locality measure $\nu$ are computed for the graph G and (2) $\mu$ is used to guide the rewiring of the edges given the density factor $\rho$ and constrained by the locality measure $\nu$. Algorithm 1 describes the computation done in step (1) and Algorithm 2 the computation in step (2).

---

**Algorithm 1 Fast $\mu, \nu$ Computation**

    **Input**: Adjacency matrix $\mathbf{A}$, Number of snapshots $L$, Connectivity radius $C = 8$
    **Output**: Locality matrix $\mathbf{D}$, Connectivity matrix $\mathbf{M}$
1: $\mathbf{D} \leftarrow \mathbf{A}$
2: $\mathbf{A}_{curr} \leftarrow \mathbf{A}$
3: $\mathbf{A}_{next} \leftarrow \mathbf{A}^2$
4: **for** $r = 2, \cdots, L$ **do**
5:     $\mathbf{A}_{curr} \leftarrow \mathrm{clip}(\mathbf{A}_{curr})$
6:     $\mathbf{A}_{next} \leftarrow \mathrm{clip}(\mathbf{A}_{next})$
7:     $\mathbf{D} \leftarrow \mathbf{D} + r(\mathbf{A}_{next} - \mathbf{A}_{curr})$              $\triangleright$ Adds nodes at distance $r$ to $\mathbf{D}$.
8:     $\mathbf{A}_{curr} \leftarrow \mathbf{A}_{next}$                           $\triangleright$ Setup next iteration.
9:     $\mathbf{A}_{next} \leftarrow \mathbf{A}_{next}\mathbf{A}$
10: **end for**
11: $\mathbf{M} \leftarrow \mathbf{A}^C$
12: **return** $\mathbf{D}, \mathbf{M}$

---

While the implementation depends on the choices for $\mu$ and $\nu$ in step (1), our specific choices are particularly efficient. Both of our measures can be in fact computed with matrix multiplication and other simple matrix operations that are highly efficient on modern hardware. For square matrices of size $n$, we set the cost of matrix multiplication to $O(n^3)$ in our analysis for simplicity although there are more efficient procedures for this operation. In particular the computational complexity – assuming the number of snapshots $L$ to be sufficiently small which is the case in practice – of $\mu$ and $\nu$ is $O(n^3)$ with $n$ being the number of nodes considered, while the memory complexity is $O(n^2)$. We further note that while computing effective resistance also has a theoretical complexity of $O(n^3)$, it involves (pseudo-)inverting the graph Laplacian which in practice has a significant prefactor when compared to matrix multiplication and often runs into numerical stability issues.

The computations in Algorithm 2 are similarly efficient and easily parallelizable as the computation for each node is independent. We find that the computational complexity is $O(n^2)$ as each node-wise calculation is $O(n)$ and there are $n$ such operations. Overall, the entire procedure of **LASER** therefore has a cost of $O(n^3)$, with the primary overhead being the cost of taking matrix powers of the adjacency matrix $\mathbf{A}$. In these calculations we are absorbing the costs due to the number of snapshots $L$ as a prefactor, which is a reasonable assumption as $L$ is constant, small, and independent of $n$. As shown in the benchmarking experiments, in practice the cubic cost shows very strong scalability as the operations required in our computations are highly optimized on modern hardware accelerators and software libraries.

---

**Algorithm 2 LASER rewiring for locality value $r$**

---

    **Input**: Graph G, Locality matrix $\mathbf{D}$, Connectivity matrix $\mathbf{M}$, Locality value $r$, Density $\rho$
    **Output**: Rewired edges $\mathsf{E}'_r$ for locality $r$
1: $\mathsf{E}'_r \leftarrow \text{empty}()$
2: **for** node in G **do**
3:     $\mathbf{D}_{node}, \mathbf{M}_{node} \leftarrow \mathbf{D}[node, :], \mathbf{M}[node, :]$
4:     $\mathbf{M}_{node} \leftarrow \mathbf{M}_{node}[\mathbf{D}_{node} = r]$              ▷ Consider only the nodes at locality value $r$.
5:     $k = \text{round}(\rho |\mathbf{M}_{node}|)$             ▷ Edges to add $k$ are the fraction $\rho$ of the orbit size.
6:     $\mathsf{E}^{node}_r \leftarrow \text{getNewEdgesRandomEquivariant}(\mathbf{M}_{node}, k)$
7:     $\mathsf{E}'_r.\text{addEdges}(\mathsf{E}^{node}_r)$
8: **end for**
9: **return** $\mathsf{E}'_r$

---

## D.2 REAL-WORLD GRAPHS SCALING

In Table 8, we show rewiring pre-processing and training times on `PCQM-Contact` and pre-processing times on `PascalVOC-SP` datasets from LRGB. `PCQM-Contact` has more than 500k graphs with an average node degree of $\approx 30$ per graph. We find that the rewiring and inference times between FOSR, SDRF, and **LASER** are very similar. `PascalVOC-SP` is a dataset with $\approx 11$k graphs, but with a considerably larger average node degree of $\approx 480$. We find that **LASER** is still extremely efficient, especially when compared to SDRF, with FOSR being the fastest. We further remark that the FOSR and SDRF implementations rely on custom CUDA kernel implementations with Numba to accelerate the computations. We specifically avoided such optimizations in order to preserve the clarity of the implementation and found the performance of **LASER** to be efficient enough without such optimizations regardless. We envision that a custom CUDA kernel implementation of **LASER** could be used for further speedups.

In Table 9, we show the training time impact of the various rewiring techniques with SAGE as the underlying MPNN on the `Peptides-func` task. We find that the rewiring techniques overall add small overhead to the SAGE implementation. In particular, **LASER** with $L = 1$ has the same training time of FOSR and SDRF but achieves significantly higher performance.

Table 8: Rewiring and Training+Inference times for the `PCQM-Contact` and `PascalVOC-SP` datasets.

| Dataset | FOSR | SDRF | **LASER** |
|---|---|---|---|
| PCQM Training+Inference Time (GIN) | 11h 47m | 11h 51m | 12h 21m |
| PCQM Rewiring Time | 5m 20s | 5m 36s | 4m 15s |
| PascalVOC-SP Rewiring Time | 4m 3s | 31m 30s | 9m 6s |

## D.3 SYNTHETIC SCALING EXPERIMENT

To further benchmark the scalability of the rewiring time, we construct an Erdos-Renyi synthetic benchmark with $n$ nodes and Bernoulli probability $p = 10/n$, i.e. in expectation we have $10n$ edges. In these synthetic experiments, we benchmark on a machine with 44 cores and 600GB of RAM. Setting $n = 10$k (meaning $\approx 100k$ edges), **LASER** ($\rho = 0.5, L = 1$) takes 11s, while FOSR and

Table 9: Run-time on `Peptides-func` with different rewirings and SAGE. For FOSR and SDRF we set the number of iterations to 40 and for **LASER** we set $\rho = 0.5$. $L$ denotes the number of rewirings.

| Model | Training Time | Rewiring Time | $L$ | Parameters | Test AP ↑ |
|---|---|---|---|---|---|
| SAGE | 46m 26s | N/A | 0 | 482k | 0.5971±0.0041 |
| SAGE-FOSR-R | 55m 51s | 35s | 1 | 494k | 0.4678±0.0068 |
| SAGE-SDRF-R | 54m 52s | 21s | 1 | 494k | 0.5892±0.0040 |
| SAGE-**LASER** | 54m 50s | 33s | 1 | 494k | 0.6442±0.0028 |
| SAGE-**LASER** | 1h 4m 28s | 42s | 3 | 495k | **0.6464**±0.0032 |

SDRF with $0.001n = 100$ iterations take 45s and 5m 48s respectively. On $n = 100$k (meaning $\approx 1$ million edges), **LASER** completes the computation in 2h 16m, while FOSR and SDRF do not terminate after more than 24 hours of computation. We emphasize that these graphs are much larger than what graph rewiring techniques are designed for, yet **LASER** is still able to handle them with success due to its efficient design.

Overall, in these synthetic experiments, we found that **LASER** is able to scale to very large graphs and with a reasonable run-time on modest hardware. Scaling to larger graphs with millions of nodes would likely require some further form of sampling, with this being a common practice used to scale GNNs to very large graphs (Hamilton et al., 2017).

