# OpenReview forum: "Locality-Aware Graph Rewiring in GNNs"
_ICLR.cc/2024/Conference — ICLR 2024 poster_

### Official Review · Reviewer_XEwE · 2023-10-25

**Soundness:** 2 fair
**Presentation:** 3 good
**Contribution:** 3 good
**Rating:** 8
**Confidence:** 3

**Summary:**

This paper focuses on graph rewiring, a technique used to reduce the over-squashing problem in GNNs. It analyzes three desiderata for graph-rewiring, pointing out that previous methods fail to satisfy all of them. Based on this, the authors propose a novel rewiring framework that satisfies all three desiderata, considering locality by a sequential process. Finally, they validate the method on various real-world benchmarks.

**Strengths:**

1. The paper clearly points out that previous graph rewiring fails to satisfy three aspects, and suggests a new method satisfying all aspects.
2. The paper is overall well written and structured.
3. The theoretical analysis provides an explanation on why the suggest method works well.

**Weaknesses:**

1. The background for preserving sparsity seems to be weaker compared to the background for preserving locality, where the authors give effective resistance as example for preserving locality. It would be more persuasive if the authors gave an experiment result or paper for the question, “some of these new connections introduced by spatial rewiring methods may be removed with affecting the improved connectivity.”
2. Authors have used “effective resistance” throughout the paper to support their claim. However, the paper does not have any comparison with the graph rewiring method that uses effective resistance$^{[1]}$.

[1] Black et al., Understanding oversquashing in gnns through the lens of effective resistance, ICML 2023

**Questions:**

1. The paper used the number of walks for the connectivity measure $\mu$. According to the preliminaries in section 2, it seems that the connectivity measure is an unnormalized adjacency matrix. However, in the theorem 4.1 of the citated paper (Di Giovanni et al., 2023), the number of walks are divided by power of minimum node degree. Following this, doesn’t the adjacency matrix should be normalized using the node degree?
2. For the necessity of sequential rewiring, authors claimed that instantaneous rewiring easily violates either locality or sparsity constraint. In Figure 3, authors conducted an ablation study on the number of snapshots. Is there any comparison with an instantaneous rewiring, i.e., snapshot being 1?
3. The connectivity and locality measures are only computed once over input graph to make the rewiring process efficient. However, it seems that these measure might change after some sequential steps of graph rewiring. For example, the shortest-walk distance between two nodes might get smaller after some graph rewiring steps, leading to another graph rewiring rather than the authors intended. Is it just that the performance gap of computing measures once and at each rewiring step did not differ much?

---

> ### Author Response · Authors · 2023-11-16
>
> We warmly thank the reviewer for pointing out that our work is clear, well written, and that it is a good contribution to the literature. We would like to address the valid points brought up by the reviewer.
>
> *Further motivating sparsity.*
>
> In LASER, the rewiring is guided through a connectivity measure that is used to prioritize the edges which would improve the connectivity the most. Consequently, LASER, given a $\rho$ factor, is “discarding” the edges that would affect the connectivity the least, according to the chosen connectivity measure.
>
> *Adding an effective resistance rewiring.*
>
> We agree with the reviewer that a baseline such as GTR (Black et al, 2023) would fit well in our comparison. We have added the baseline to our experiments, alongside the inclusion of another recent curvature rewiring technique BOFR (Nguyen et al., 2023), as requested by another reviewer. The two are the most recent state-of-the-art rewiring techniques. In Tables 2 and 3, we demonstrate that LASER shows strong performance over these two new baselines as well.
>
> *Number of walks in connectivity measure.*
>
> We thank the reviewer for the interesting question. We would like to clarify that the node degrees in the bound derived in Proposition 5.1 come from the convolutional part of the GNN model. It is standard practice to normalize such a convolution operation for training stability reasons. Instead, in our connectivity measure $\mu$ we calculate the number of walks from node $u$ to $v$ of length $k$, which requires taking powers of an unormalized adjacency matrix. The two operations are therefore unrelated to each-other.
>
> *Instantaneous rewiring.*
>
> In previous works such as FOSR and GTR, the authors have shown experimentally that relational types tend to significantly improve the performance of rewiring techniques. As such, we have not included such an ablation study. This can also be motivated from the fact that relational models are always able to recover “instantaneous” models by setting the weight matrices to be equal to each-other. For this reason, they may be considered as strictly more powerful instances than instantaneous models.
>
>
> *Efficient computation of the connectivity and locality measures.*
>
> It is indeed true that the sequential snapshots change the topology of the graph which would affect the number of walks. Computing the connectivity measure once, as you have pointed out, gives an important computational speed-up which we wanted to keep for scalability reasons. During the development of the model, we found that such a choice did not have a noticeable impact on the performance
>
> We once again thank the reviewer for supporting our work and for the excellent questions. We are of course happy to keep engaging throughout the rebuttal period.

---

> > ### Comment · Reviewer_XEwE · 2023-11-22
> >
> > I thank the authors for their comprehensive response, and have no further questions.

---

### Official Review · Reviewer_KgxZ · 2023-10-27

**Soundness:** 2 fair
**Presentation:** 3 good
**Contribution:** 2 fair
**Rating:** 5
**Confidence:** 4

**Summary:**

This paper presents a framework for graph rewiring. Specifically, the paper suggests having two competing metric - one for oversquashing and one for distance to the original graph that should be balanced. The paper presents a specific instantiation of the framework and present numerical experiments to show the benefit of the framework.

**Strengths:**

**Originality**

The framework presented in the paper is new. However, the idea of preserving the original graph structure is not new, further as the authors themselves state the idea of using relational GNNs is not new either. Though they do have an original extension of the framework.

**Quality**

Please see weaknesses.

**Clarity**

The paper is well written and places itself very well in the context of prior work. This I think is the papers biggest strength. The framework is clearly presented. However, the paper's clarity degrades in Section 5. For example Proposition 5.2 is not clear and there is no formal statement or proof anywhere in the paper.

**Significance**

The paper's method does seem to perform better than SDRF and FOSR. However, I have certain concerns about the experiments, highlighted in the weakness section.

**Weaknesses:**

I think there a few weaknesses. Part of my concern with the paper is that as detailed in point 2a, there are now many different rewiring techniques. However, I do not think we understand oversquashing yet. Hence for me new papers in the area either need to a thorough comparison with prior work to show empirical improvement. Or contributes to understanding oversquashing and I think this paper, unfortunately, does not do either.

1) One big weakness is Proposition 5.2. The statement in the paper is informal and incomplete. However, the paper does not have a formal statement or a proof. This is a big concern. For example, for the informal statement $\mathcal{G} = \emptyset$ vacuously gives us the result. However, that version is meaningless. Hence a formal statement is needed. There is space in the paper for this discussion, the first few pages are quite repetitive.

I see that page 14 has something that is called a proof. But without a formal statement the notion of proof does not make sense to me. And there is no formal statement. A.2 is thought of to be formal statement but it refers to a lower bound in 5.2 which is not clear.

2) I have a few concerns about the experiments.

    a) First, I think the paper compares against very few prior works. The paper does a good job of citing many prior works in the area but then only compares against two of them. The paper should compare against most of the following or explain why it is not relevant to do so: GTR (Black et al. 2023), BORF (Nguyen et al. 2023), DRew (Gutteridge et al 2023), DIGL (Gasteiger et al 2019), Expander propagation (Deac et al 2022), DiffWire (Arnaiz Rodriguez et al 2022). The paper even cites Brüel-Gabrielsson et al., 2022; Abboud et al., 2022, and  Banerjee et al., 2022 as further works with rewiring techniques.

   b) I also have some concerns with the experiments that are present. First, as the paper notes the network from FOSR is the case that $L=1$. However, for the method proposed in the paper, the paper uses $L \in \{2,3,4,5\}$. Since for each $\ell \le L$ we have a different weight matrix, this implies that the networks for LASER are bigger than the networks used for FOSR and SDRF. This is an inequity that could account for the improved performance.

   c) Hyperparameter tuning. The paper mentions that they tune $L$ and $\rho$ for their method. However, they do not perform any hyper parameter tuning for the comparison methods (SDRF and FOSR). They fix the number of edges to be 40. This is another thing that could account for the inequity between the methods. It is also not mentioned what number is reported, I am assuming that the experiments trained models for each of the hyperparameters, picked the setting with the best validation performance and then reported the test error, however it would good if this were explicitly mentioned (since Figure 3 reports the metrics on the test data for all hyperparameters).

3) I think the fairer version for the experiments would be to sequentially rewire the datasets with FOSR and SDRF such for each $1 \le \ell \le L$ all sets $E_\ell$ have the same size. I think this would help determine if part of the reason for increased performance is the rewiring or the new GNN architecture.

The next couple of concerns are more minor.

4) In terms of the context for the work, I think the following could be clarified. The notion of oversquashing in Alan and Yahav 2021, and the other papers Topping et al 2022, Black et al., 2023; Di Giovanni et al., 2023 are subtly but in my opinion importantly different. Alon and Yahav 2021, is a more information theoretic issue that is highlighted. Vectors of a certain size can not aggregate too much information. However, the issue in Topping et al 2022, Black et al., 2023; Di Giovanni et al., 2023 is more about optimization rather than information theory. These papers talk about how the Jacobian has a small norm. Hence while both phenomena can be labelled as oversquahing I do think they are different and should be treated as such.

5) The paper measures the "information" provided by the graph structure as preserving locality. Specifically, they say ``while the measure $\nu$ is any quantity that penalizes interactions among nodes that are ‘distant’ according to some metric on the input graph.'' However, local structure of the graph and the information stored in the graph are not the same thing.

**Questions:**

See weaknesses

---

> ### Author Response · Authors · 2023-11-16
>
> We sincerely thank the reviewer for the excellent and thorough review. We would like to address the points in full.
>
> *Regarding Proposition 5.2.*
>
> We thank the reviewer for the feedback on the Proposition. We have followed the suggestion and replaced the informal statement with the precise one from the Appendix. To make things more concrete, we have formulated the result for the case of a graph (pointing out that the size can be taken to be positive to avoid vacuous edge cases) and rewritten the proof in the Appendix (Section A) more explicitly. The discussion surrounding Proposition 5.2 has also been modified in order to improve clarity. Thanks again for the feedback.
>
> *Further baselines.*
>
> We thank the reviewer for the suggestions. We agree that the paper would benefit from further comparisons. Accordingly, we have added a comparison to GTR (Black et al. 2023) and BORF (Nguyen et al. 2023) in our work. We believe these to be the most direct and relevant recent comparisons to our framework, as these works aim to modify the computational graph as a pre-processing.
>
>
> *Conservation of parameters and tuning.*
>
> We thank the reviewer for the valid experimental doubts, we would like to address these in full. All experiments in our work *strictly follow the 500k parameter budget limit* for LRGB. As for the TUDatasets, we keep the hidden dimension fixed to 64. Furthermore, in the TUDatasets we do not tune on more than 3 snapshots (details in the Appendix). As such the model parameter count remains extremely similar, especially when compared to the relational FOSR, GTR, BOFR, and SDRF models. Therefore, we are certain that the slight increase in parameters does not affect our experimental evaluation.
>
> We agree with the reviewer that fixing the iterations in SDRF and FOSR to 40 may have favoured LASER in our previous evaluation. Therefore, we have included results achieved with a more thorough sweep of the hyper-parameters for SDRF and FOSR, alongside the additional GTR and BOFR benchmarks. We have added detailed information in the Appendix (Section B) on the parameters searched.
>
> *New experiment proposed.*
>
> We thank the reviewer for the suggestion. Having a sequence of rewirings is a contribution of our work and it is not immediately clear how easily this translates to other rewiring techniques. We agree, however, that this is an interesting experiment and show in the Appendix (Section C.4) that additional snapshots even slightly decrease the performance with FOSR, while tend to improve the performance with LASER.
>
> *Minor concern on the literature summary.*
>
> This is an accurate description of the literature landscape thus far. We fully agree with your point of view, even though we may have summarized some of the references in the interest of space. If you believe there are certain points where it is worth making such clarification explicit, we would be happy to add it.
>
>
> *Minor concern on the locality measure.*
>
> Thank you for this observation. We agree that we could make our point of view more explicit. Rather than “information”, preserving locality in our work mainly refers to preserving the inductive bias afforded by the input graph topology, according to which features that are associated with nearby nodes should interact “more easily” or “more often” than those associated with distant nodes. Naturally, we are tacitly assuming here that the graph-structure we are given in the first place is somewhat aligned with the downstream task in order for this claim to be valid. This is why “some metric”, mentioned above, should capture a notion of distance on the graph that we believe to be, in most cases, aligned with the task. We have rephrased a few points (see e.g. the paragraph `Instantaneous vs Sequential Rewiring’ or the discussion above the new statement of Proposition 5.2) of the submission to emphasize that “locality” is a “bias” we aim to preserve in most tasks we believe GNNs to be useful for.
>
> We again thank the reviewer for the great questions. We hope that with our rebuttal and the improvements we have made to the paper, we have addressed the concerns found regarding the proposition and the experiments, and that this may convince the reviewer to increase the score. We are of course very happy to keep engaging during the rebuttal period.

---

> > ### Comment · Reviewer_KgxZ · 2023-11-17
> > **Proposition 5.2**
> >
> > I thank the authors for responding to my complaints. While I am still looking at the new experimental changes, I still have concerns about Proposition 5.2
> >
> > For the current version, doesn't picking $\rho=0$ make it vacuously true again?
> >
> > I tried reading the argument again, to see if I could extract what the correct statement might be, but I couldn't. I produce it here with my concerns. **Please let me know how to resolve these.**
> >
> > ----------------
> > Consider a family of graphs $G$ and a spectral-rewiring approach $\mathcal{R}$ that is based on adding edges based on those that maximally decrease the effective resistance. For simplicity, say that this method only adds a single edge. We let $\gamma > 0$ be the lower bound of maximal pairwise effective resistance over all graphs in $G$.
> >
> > **First complaint, the new statement does not reference a family of graphs. However, I am okay letting $\mathcal{G} = \{G\}$.**
> >
> > Since the pairwise distance is always larger or equal than the effective resistance, **(Is there a reference for this?)** we can bound the Frobenius norm of the difference between the distance matrices as
> >
> > $\|D\_G−D\_{R(G)}\|_F \ge |\gamma−1|.$
> >
> >
> > For the LASER method instead, where we have $L$ snapshots and a percentage of sampled edges equals to $\rho$, we can estimate the Frobenius norm as:
> >
> > $\|D\_G - D\_{LASER}\| = \sqrt{\sum\_{\ell=1}^L \ldots \ell^2}$.
> >
> > **Second complaint: I don't see how this equality is true. the terms here are for the change to the distance for the endpoints of the edges added. However, adding an edge between $u$ and $v$ doesn't only change the distance between $u$ and $v$. It could change the distance between other nodes. Further, the successive adding of edges could alter prior distances as well. Hence I ask the authors for more details on this.**
> >
> > Assume now that the family of graphs is sparse **(This is not in the statement of the proposition or resolved later)** with most nodes having small degree (as for molecular graphs) **(This is also not in the statement of the proposition or resolved later)** and that $\rho$ **(Like $\rho = 0$?)** is small, then we can approximate the lower bound as above with some constant $c$
> > as
> >
> > $\|D_G - D_{laser}\| \ge \text{some expression} \le \sqrt{ncL^3}$
> >
> > **Third complaint: the inequality at the ends go in opposite directions. So tell us nothing about $\|D_G - D_{laser}\|$ vs $\sqrt{ncL^3}$. Then the proof continues making some assumptions about $L$, etc until it gets to the following**
> >
> > Now, choose ρ such that then
> >
> > $\gamma > 1 + \sqrt{ncL^3}$
> >
> > **Fourth complaint: We only assumed that $\gamma > 0$, however this says $\gamma > 1$. How do we know that?**
> >
> > **Fifth complaint: Assuming the rest is true, I imagine the rest of the proof is as follows**
> >
> > $\gamma > 1 + \sqrt{ncL^3} \Rightarrow \gamma - 1 >  \sqrt{ncL^3} $, then want to say $\sqrt{ncL^3} \ge \|D_G - D_{LASER}\|$ but as pointed out above this is not known.
> >
> > **Hence not sure how the conclusion is reached.**

---

> > > ### Comment · Reviewer_KgxZ · 2023-11-17
> > > **Experimental Changes**
> > >
> > > I thank the authors for the changes and their patience and engagement. I do believe that experiments are better, but I have a few questions.
> > >
> > > 1. Could the author present the sizes of the networks?
> > > 2. The paper seems to no longer report the performance for the relation version Table 3.
> > > 3. The numbers in Table 3 seem different from before (for some columns like Reddit Binary, but basically unchanged for other ones). Is this due to the randomness in the data split?

---

> > > > ### Author Response · Authors · 2023-11-18
> > > >
> > > > We thank you for the experimental suggestions and are happy to hear that you believe that the section is now improved. We would like to answer your new questions below:
> > > >
> > > > > Could the author present the sizes of the networks?
> > > >
> > > > - For the LRGB datasets, the networks all sit very close to the 500k parameter budget.
> > > >
> > > > - For the TUDatasets, the “single snapshot” models have 13254 parameters. This includes the GCN without rewiring and FOSR, SDRF, GTR, and BORF operating without a relational model.
> > > >
> > > > - Models with two snapshots, including R-FOSR, R-SDRF, R-GTR, R-BORF and LASER all have 17350 parameters.
> > > >
> > > > - Finally, the highest snapshot version of LASER used for the TUDatasets, with 3 snapshots, has 21446 parameters.
> > > >
> > > > These model sizes are all achieved by fixing the hidden dimension to 64, as done in previous works.
> > > >
> > > > > The paper seems to no longer report the performance for the relation version Table 3.
> > > >
> > > > For clarity and space, we have merged the results for the relational and non-relational baselines into a single entry. This means that, for example, the entry FOSR includes tuning over both a relational and non-relational version of FOSR (effectively treating the relational component as a hyper-parameter). We believe that keeping the two separated does not add to the evaluation and may in fact make comparisons more challenging due to the doubled number of baselines. Of course, as in all of our experiments, we report the test score of the model that achieved the highest validation score. We have now made this more clear in the work, and thank the author for highlighting any potential confusion.
> > > >
> > > > >The numbers in Table 3 seem different from before (for some columns like Reddit Binary, but basically unchanged for other ones). Is this due to the randomness in the data split?
> > > >
> > > > The numbers changed as we have merged the relational and non-relational models and we have also tuned them over a more fair hyper-parameter sweep, as detailed in the Appendix. What we report is the test score of the model that achieved the highest validation score. This means that some entries remained the same, while others changed as a new hyper-parameter configuration achieved a higher validation score due to the more extensive hyper-parameter tuning.  You are right in pointing out that the stochasticity due to the data split also plays a role in this.
> > > >
> > > > We are very happy to keep engaging with any potential further questions. We once again thank you for the excellent comments and for the time you have dedicated to our work.

---

> > > > > ### Comment · Reviewer_KgxZ · 2023-11-20
> > > > > **Thank you**
> > > > >
> > > > > I thank the authors for their responses. I still have some concerns in relation to the experiments. However, with the addressed concerns. I increase my score to a 5.

---

> > > > > > ### Author Response · Authors · 2023-11-20
> > > > > >
> > > > > > We sincerely appreciate your effort and engagement during the rebuttal period. We are more than happy to address any remaining experimental concerns that could strengthen our work further. We believe that your experimental suggestions, especially regarding W2 and W3, have helped to significantly enhance our experimental section, and we look forward to hearing how you believe we can further improve our work.

---

> > > > > > > ### Author Response · Authors · 2023-11-22
> > > > > > >
> > > > > > > We would like to kindly inform you about the newly added baseline, DiffWire, and the additional tuning we have done for **LASER**, both of which can be found in the newest version of the manuscript. We would like to point to the recent discussion with reviewer RVSP for further details. We finally thank you once again for your efforts during the rebuttal.

---

> > > ### Author Response · Authors · 2023-11-18
> > >
> > > We thank the reviewer for the detailed analysis and excellent questions – we genuinely appreciate the time spent to review our paper. Unfortunately, we agree that we have been a little hasty with the upper bound on LASER since clearly we were ignoring contributions to other pairwise distances, so we thank the reviewer for spotting that. While all the other bits in the argument would work, upon closer inspection we have realized that finding a general bound for the Frobenius norm of the difference between distance matrices when using **LASER** is, in fact, all but trivial. Regardless, we raise here two important points which we believe still heavily support our original claim:
> > >
> > > - In many real-world scenarios, adding edges between nodes that are closer in space is still going to affect the norm above less than adding edges between distant nodes so as to maximally decrease the effective resistance. Note that we had already validated this claim in practice by reporting the norm of the difference of the distance matrices for the more than 15k real-world graphs from the Peptides dataset in Figure 2.
> > >
> > > - To analyse a concrete case, we have presented a new statement  (the new Prop. 5.2) where we have adapted the argument to the case of a “lollipop” graph, that represents a prototypical graph with `bottleneck’; in fact, the analyses in Di Giovanni et al. (ICML 2023) and Black et al. guarantee that this is one of the worst-case scenarios regarding the commute time and hence over-squashing. We have phrased the statement explicitly and adapted the discussion surrounding it. Note that in the statement we can take $\rho > 0$ and in the proof we have explicitly given a bound $0 < k < \sqrt{L}/2$ on the number of edges $k$ one can choose to add with **LASER**.  The newly written section alongside the proof in the Appendix can be found in the most recent version of our uploaded manuscript.
> > >
> > > Thank you again for your careful analysis and for spotting our mistake. While we have modified Prop 5.2 to account for a specific “bottleneck” case, we believe that we have shown experimentally that the general intuition that LASER is more local than spectral rewirings is valid. A general statement comparing spatial and spectral rewiring might be significantly more challenging to be achieved, and most likely will require a rethinking of the characterization of locality and some further assumptions. Nonetheless, the main goals of the paper consisted of proposing a general framework for rewiring that could both combine spatial and spectral principles, along with an extensive experimental evaluation against recent geometric (curvature-based) and spectral rewiring baselines. We are very thankful for your time and effort in evaluating our work.

---

### Official Review · Reviewer_RVSP · 2023-10-31

**Soundness:** 2 fair
**Presentation:** 3 good
**Contribution:** 2 fair
**Rating:** 3
**Confidence:** 3

**Summary:**

This paper explores the concept of graph rewiring, which involves altering graph connectivity to improve information flow. Three essential objectives for graph rewiring are identified: (i) reducing over-squashing, (ii) preserving the graph's local structure, and (iii) maintaining its sparsity.

The authors highlight that there is a trade-off between two primary techniques in graph rewiring: spatial and spectral methods. They argue that spatial methods tend to address over-squashing and local structure but may not preserve sparsity, while spectral methods generally handle over-squashing and sparsity but might not maintain local properties.

To tackle these trade-offs, the paper introduces a novel rewiring framework. This framework employs a sequence of operations that are sensitive to the graph's local characteristics, aiming to simultaneously meet all three objectives: reducing over-squashing, respecting the graph's locality, and preserving its sparsity. Furthermore, the paper discusses a specific instance of this proposed framework and evaluates its performance on real-world benchmarks.

**Strengths:**

The authors gave a nice taxonomy of rewiring methods and the issues that they suffer from

**Weaknesses:**

* In the paper they constantly cite spectral methods such as Arnaiz-Rodríguez et al., Black et al., 2023 and transformer-based methods such as Kreuzer et al., 2021; Mialon et al., 2021; Ying et al., 2021; Rampasek et al., 2022. But in the tables, there is no comparison with these methods.

* The results are very poor, especially when it comes to the task of graph classification, where the method is not able to outperform the few selected models.

**Questions:**

* Why do you say that spectral methods are not local, since most of them combine long-range information (by bypassing the bottleneck) and initial neighborhood? For instance, Arnaiz-Rodríguez et al. combine the CT model (long-range) with MPNN with the initial adj.

* Is there any study of the parameter k? For instance, what happens for large k values? How does it affect the relationship between the distance of nodes of the same cluster (locally) and nodes of different clusters (globally)?

* Can there be any attention mechanism between the snapshots, attending to those snapshots that contribute more to the representation of the graph, as they do in multiple papers where they explore different adj  (Abu-El-Haija et al., 2019; or FSGNN (Improving Graph Neural Networks with Simple Architecture Design)?

---

> ### Author Response · Authors · 2023-11-16
>
> We thank the reviewer for the excellent comments and questions, below we would like to address them in full.
>
> *Other baselines.*
>
> We thank the reviewer for pointing out further baselines. We agree that the work would be strengthened via the addition of more baselines. Accordingly, we have added to our experiments a comparison to GTR (Black et al, 2023), and BORF (Nguyen et al, 2023).  We believe that these techniques are the most comparable to ours as they are designed to augment the graph through pre-processing. Transformer based methods are not directly comparable to our technique as they require significantly more computation. In fact, even the most recent state-of-the-art rewiring techniques GTR and BORF do not compare with Transformers-based architectures in their evaluation and compare against the same benchmarks present in our own work.
>
> *Poor results.*
>
> We politely disagree with the statement and believe the results to be validating our point. We kindly ask for further clarification as to why you believe the results to be poor. In fact, when it comes to comparing with similar methods that also aim at adding edges more “surgically”, the LASER framework consistently outperforms or is on par with the other baselines in the worst-case scenario. Even with the new added baselines, this claim still holds. More concretely, in Table 2 LASER beats all baselines consistently. In Table 3, LASER achieves a mean rank of 1.67, with the second best model achieving a mean rank of 3.20.
>
> *Spectral methods are not local.*
>
> We thank the reviewer for the interesting question. We would like to clarify that in our work spectral rewirings refer to methods that aim to rewire the graph based on spectral quantities, which is not a local mechanism. This is because rewiring through quantities such as increasing spectral gap or decreasing commute time accounts for the global topology of the graph, meaning that the edges to be added end up minimizing spectral (not spatial) quantities and hence may, quite often, violate any spatial constraint. Put differently, the edges added through such methods will tend to be very non-local. This is something that has also been pointed out in Di Giovanni et al., ICML 2023. Architectures such as DiffWire aim to address this problem by adding additional components to the model. We emphasize that, instead, in our work, as in FOSR, SDRF, GTR, and  BORF, the goal is to directly modify the computational graph of any existing MPNN, instead of proposing a completely different architecture.
>
> *Is there any study through parameter k?*
>
> If this question is referring to parameter $k$ in Equation 8, then this parameter simply indicates that we want to consider walks of length $k$ on our graph as our connectivity measure. In general, we want $k > L$ and we also set $k$ to be a power of 2 as this leads to more efficient matrix multiplication algorithms due to efficient factorization of the multiplication. As such, we fix $k=8$ in our work as a tradeoff between $k=4$ which would be too close to $L$ and $k=16$ which would be unnecessarily large. We found the network to be rather insensitive to this choice as long as $k$ was large enough.
>
> *Can there be an attention mechanism between the snapshots.*
>
> We thank the reviewer for the suggestion. We agree that this is something that could definitely be a future direction of our snapshot framework. It is indeed natural to explore how to best aggregate different snapshots together. In our work, we take an approach inspired by Temporal GNNs, but other approaches such as attention mechanisms, could be explored in future work.
>
> We thank once again the reviewer for the comments and hope that we have clarified doubts in the new version of the paper, with a significantly stronger experimental section. We hope that this is able to convince the reviewer to increase the score. We are of course happy to keep engaging with the reviewer.

---

> > ### Comment · Reviewer_RVSP · 2023-11-21
> > **Comment to authors.**
> >
> > Sorry but the experimental results are still inconclusive. The new baseline BORF is still better in MUTAG and GTR is better in REDDIT.
> >
> > Existing methods such as Diffwire are cited and considered in "Future Work". The authors say: In this paper, we considered a simple instance of the general rewiring paradigm outlined in Section 4, but we believe that an interesting research direction would
> > be to explore alternative choices for both the connectivity and locality measures, ideally incorporating
> > features in a differentiable pipeline.

---

> > > ### Author Response · Authors · 2023-11-21
> > >
> > > We thank you for the engagement with our work and for your effort during the rebuttal. We truly believe that your suggestions have helped to strengthen our work. We would like to address your points in full.
> > >
> > > We highlight that **LASER** achieves the top performance on **6 out of the 9** datasets we evaluate on, when **compared against 5 baselines**. This is in very stark contrast to any rewiring technique we evaluate against. Further, the datasets on which **LASER** does not achieve first place coming in as a close competitor are *problematic datasets*. We point out the dataset flaws in the Appendix, but we included such tasks regardless for completeness of comparison as all the existing rewiring techniques evaluate almost exclusively on them. To be more specific: MUTAG *only contains 18 graphs in the test set*, while REDDIT-BINARY and IMDB-BINARY *do not have node features*. We refer to the Appendix (Section B) for a more detailed discussion. Having said this, **LASER** still achieves competitive performance on them, regardless. Of course, the quirks of these tasks mean that they are more prone to random fluctuations in the results, with **LASER** still remaining within a standard deviation of first place.  We would like to highlight that we focus most of the ablations on the higher quality LRGB benchmarks, as again we have found flaws in the benchmarking of existing rewiring techniques evaluating solely on the TUDatasets. On the higher quality LRGB tasks, **LASER** is the clear favourite.
> > >
> > > For these reasons, calling the results *inconclusive* and *very poor* seems rather unjustified, again as **LASER** ranks first on 6 out of 9 of the tasks we consider.
> > >
> > > Regarding, DiffWire, we would like to highlight that the technique operates under a different regime to ours by proposing an entirely different GNN architecture. Instead, in our work, as in FOSR, SDRF, BORF, and GTR, we simply aim to augment *any MPNN* through a rewiring technique as a pre-processing step. We would like to point out that the most recently accepted works BORF and GTR *also do not evaluate against DiffWire*, again because the technique is not directly comparable. Furthermore, even in these works the techniques do not achieve the highest accuracy on every task in their experimental section.
> > >
> > > For these reasons, we hope that you consider increasing your score. We truly believe we have strengthened our experimental section by providing 2 additional baselines throughout the rebuttal, following your helpful suggestions. We finally sincerely thank you once again for your engagement during the rebuttal.

---

> > > > ### Comment · Reviewer_RVSP · 2023-11-21
> > > > **Response to authors**
> > > >
> > > > **Comment 1** We highlight that LASER achieves the top performance on 6 out of the 9 datasets we evaluate on, when compared against 5 baselines. This is in very stark contrast to any rewiring technique we evaluate against.
> > > >
> > > > A look to table 3:
> > > >
> > > > REDDIT-BINARY :  LASER is 3rd (competitive)
> > > > IMDB-BINARY :     LASER is 2nd wrt NO REWIRING (-2%)
> > > > MUTAG:                 LASER is  2nd wrt BORF (-3%)
> > > > ENZYMES:             LASER is 1st
> > > > PROTEINS:            LASER is 1st
> > > > COLLAB:                LASER is 1st
> > > >
> > > > So in 6 datasets it wins on 3 and competitive in 1. I do not consider mine a "stark" judgement. In graph classification, one uses the degree if no features are available. The fact that the method works no so well in non-featured graphs means that rewiring is not so topological as explained in the exposition of the technique.
> > > >
> > > > **Comment 2** Regarding, DiffWire, we would like to highlight that the technique operates under a **different regime** to ours by proposing an entirely different GNN architecture. Instead, in our work, as in FOSR, SDRF, BORF, and GTR, we simply aim to augment any MPNN through a rewiring technique as a pre-processing step. We would like to point out that the most recently accepted works BORF and GTR also do not evaluate against DiffWire, again because the technique is not directly comparable. Furthermore, even in these works the techniques do not achieve the highest accuracy on every task in their experimental section
> > > >
> > > > What is a different regime? What if the GNN architecture is different? It is in the state of the art or not. Why exclude this technique if code is available? As far as I know, it is inductive and generalizes from seen graphs. If Diffwire is not so good enough, why do not exploit that fact? BTW, have you made a double check of comparing against a simple KNN rewiring? Of course, KNN rewring is not top techique but gives valuable information.

---

> ### Author Response · Authors · 2023-11-22
>
> We would like to thank you again for engaging with our work and for providing valuable feedback.
>
> Following your suggestions, we have added to Table 3 DiffWire as a baseline. Furthermore, we ran a full grid search for **LASER** over the number of snapshots $\\{ 2, 3 \\}$ and $\rho$ in $\\{0.1, 0.25, 0.5\\}$. We highlight that previously we had only used a random subset of $3$ configurations. We emphasize that this is a grid search of size $6$, which is now the same grid search size used for FOSR, SDRF, GTR, and BORF. For fairness and to avoid snooping, we ran this grid search *on all the TUDataset tasks* for **LASER**, explaining why most columns changed for **LASER**. We provide additional hyper-parameter details in the Appendix (Section B). We report the improved results in the newly uploaded PDF.
>
> We sincerely hope that with these additional results, we have addressed your remaining concerns and that this may convince you to raise your score. We would like to once again thank you for your efforts in reviewing our work. We are happy to answer any further questions.

---

### Official Review · Reviewer_EJ4V · 2023-10-31

**Soundness:** 3 good
**Presentation:** 3 good
**Contribution:** 3 good
**Rating:** 6
**Confidence:** 4

**Summary:**

This paper proposed a sequential rewiring method, LASER, that improves connectivity, and preserves locality in the original graph, and theoretically alleviates the over-squashing problem. Empirical experiments show that LASER outperforms the baselines on some LRGB and TUDatasets.

**Strengths:**

- This paper gives a good summarization of spectral and spatial rewiring methods.
- The anti-over-squashing and sparsity motivation of the paper makes sense.
- The sequential rewiring idea is pretty good so that edges are not added at once but more carefully selected.
- The connection between sequential rewiring and multi-relational GNN is novel
- The writing is good and clear.

**Weaknesses:**

- The paper does not explain why adding distant edges is not a good choice. In other words, why must we respect the locality and inductive bias of the given graph? Therefore, the paper does not fully convince me of their significance, although the sparsity motivation is good, the method is novel, and the experimental results seem good.
- The paper selects some spectral rewiring baselines but does not compare with DRew [1] and SP-MPNN [2] in the experiments, which also attend multi-hop neighbors in their message passing scheme and should be considered as spatial rewiring baselines. On LRGB datasets, DRew seems even better.
- As this is a rewiring approach, it does not seem to make sense to do experiments with PCQM-Contact, which is an edge prediction task
- The choice of connectivity measure in equation (8) is not efficient. The matrix multiplication would also be O(N^3). If the matrices are sparse, then the complexity would be at least O(N^2 * d_max).

[1] Gutteridge, Benjamin, Xiaowen Dong, Michael M. Bronstein, and Francesco Di Giovanni. "Drew: Dynamically rewired message passing with delay." In International Conference on Machine Learning, pp. 12252-12267. PMLR, 2023.

[2] Abboud, Ralph, Radoslav Dimitrov, and Ismail Ilkan Ceylan. "Shortest path networks for graph property prediction." In Learning on Graphs Conference, pp. 5-1. PMLR, 2022.

**Questions:**

On the top of page 3, what does the notation 2|E|R(v, u) stand for?

---

> ### Author Response · Authors · 2023-11-16
>
> We are happy to hear that the reviewer found our work to be well-motivated and our ideas novel. Below we answer the questions posed by the reviewer, which we believe have helped to strengthen our work.
>
> *Why must we respect locality?*
>
> We thank the reviewer for the important question. Preserving locality is a natural inductive bias to have whenever we assume that the graph-structure associated with the data is aligned with the downstream task.
> For instance, molecular systems observe long-range interactions that decay with the distance, in the form of Coulomb electrostatic forces. This  behaviour also naturally appears in social networks, transaction networks, or more generally in physical systems, in which interactions that are nearby are more likely to be important for a given task. Accordingly, given a budget of edges to be added, it is sensible to prioritise adding connections between nodes that are closer.
>
> To make this intuition more concrete, in Proposition 5.2 we have argued that methods that do not preserve the locality bias, may alter the distance matrix associated with the graph quite more significantly.  We have added a detailed section in the Appendix (Section C.4) alongside a pointer within the main section to tackle this point more extensively within the paper as we agree that this is an important aspect of our work. In this new section, we touch upon concrete examples, and provide a further ablation that supports this view.
>
> *Other rewirings (DRew and SP-MPNN).*
>
> We would like to point out that SP-MPNN can be seen as a special case of LASER where we take $\rho=1$ and force the weight matrices associated with each hop to simply be a convex combination of learnable coefficients. The advantage of our approach, which extends SP-MPNN, is that it provides a tradeoff between performance and efficiency since we can more easily control the sparsity of the computational graph through the factor $\rho$. Concerning the framework of DRew, we highlight that the main purpose of that work is showing that introducing delay can be particularly beneficial when exploring deep models. Because of that, their evaluation on LRGB does not stick to the convention of 5-layers architecture – which instead we have adopted here. Considering that the main goal of this project is showing how to condition spatial-rewiring on more global (spectral) properties, we believe that focusing on spectral and curvature-based rewiring baselines is fair. To strengthen our comparison to existing state-of-the-art baselines that follow our *exact* paradigm, we have added an additional spectral baseline (GTR, Black et al. 2023) and an additional curvature baseline (BORF, Nguyen et al. 2023) to our experiments, as suggested by some of the other reviewers.
>
> *Edge prediction.*
>
> We agree that edge prediction is a less natural task for rewiring techniques but included it regardless as an interesting experiment to cover the different types of tasks GNNs are used for (graph-level, node-level, and edge-level predictions).
>
> *Scalability of LASER.*
>
> While matrix multiplication has cubic run-time, it is a highly parallel operation that scales extremely well on modern hardware. It is numerically more stable than matrix inversion (used for effective resistance based rewirings such as GTR, which is also cubic), and significantly cheaper than curvature computation (used by SDRF and BORF). We show in the Appendix (Section D) that we are able to scale easily to graphs of *100k nodes and ~1 million edges*, unlike SDRF and FOSR. We point out that the other rewiring operations rely on optimization libraries such as Numba and Multiprocessing, while we avoid such an approach as we found LASER to be extremely efficient and scalable regardless. We also finally emphasize that such an operation is done only once as a pre-processing step.
>
> *What does 2|E|R(v, u) stand for?*
>
> Thanks for this question, we will clarify the notation in the text. |E| is the number of edges and R(v, u) is the effective resistance between nodes v and u. The equation CT(v, u) = 2|E|R(v, u) is a well known result and relates the commute time between v and u CT(v, u) to the effective resistance between v and u R(v, u).
>
> Once again we thank the reviewer for the excellent questions and hope that our reply persuades the reviewer to increase the score. We are of course happy to keep engaging with the reviewer throughout the process.

---

> > ### Comment · Reviewer_EJ4V · 2023-11-17
> > **Post rebuttal**
> >
> > Thank you for your detailed rebuttal, I increased my score.

---

> > > ### Author Response · Authors · 2023-11-20
> > >
> > > We would like to thank you again for your valuable feedback in your original review and for raising your score. We believe your questions and comments have helped strengthen our manuscript. Please let us know if you have suggestions to further improve our work.

---

### Author Response · Authors · 2023-11-16

We thank the reviewers for having taken the time to read our work and for the fruitful questions and comments. We truly believe that they have helped to strengthen the paper. We are particularly happy to see that the paper’s quality has been appreciated by all reviewers. We address here important points that have appeared repeatedly between reviewers and summarise the list of improvements we have made to the submission.

*Additional baselines.*

We agree with reviewers that the work would benefit from a comparison to additional baselines. In our rebuttal, as suggested, we have added the two most recent state-of-the-art graph rewiring that appeared at ICML 2023: an effective resistance rewiring GTR (Black et al, 2023), and a curvature rewiring BORF (Nguyen et al, 2023). We have also swept over a larger number of rewiring settings for the rewiring baselines (GTR, BORF, SDRF, and FOSR). Whilst doing so, we respected the parameter limitations introduced by previous works. We have added further details in the Appendix (Section B) on the hyper-parameters searched.

Overall, we found that LASER greatly outperforms the rewiring baselines on the LRGB datasets and performs very well on the existing TUDataset benchmarks used in the previous rewiring works (GTR, BORF, SDRF, and FOSR). We believe that these existing baselines further strengthen the experimental evaluation of our work. We further believe that we now have a more extensive evaluation when compared to existing rewiring works that mostly focus their evaluation solely on the TUDatasets, of which we point out some potential flaws in our Appendix (Section B.2).

*Better motivating locality.*

Some reviewers wished for a more clear motivation for locality. We agree that this is something that could be emphasized more in our work. Accordingly, we have added a detailed section in the Appendix (Section B.4) alongside a pointer within the main section to tackle this point more thoroughly within the paper. We also added an additional ablation in Section B.4 to support our claims experimentally.

We believe that our improved experimental section further supports the locality desideratum. For instance, spectral non-local rewirings based on effective resistance (GTR) or spectral gap (FOSR) perform consistently worse than LASER on the chemical tasks in LRGB. In general, preserving locality is a natural inductive bias to have whenever we assume that the graph structure associated with the data is aligned with the downstream task, which is a general assumption when using GNNs. In Section B.2, we further argue that an important motivation of locality is that it helps with generalization over graphs of different sizes, something which the current spectral rewirings potentially struggle with.

We summarize the changes we have made to our newly uploaded submission:

- Added GTR and BORF baselines to Table 2 (LRGB).
- Added GTR and BORF baselines to Table 3 (TUDatasets).
- Swept over a larger amount of hyper-parameters for the existing rewiring techniques.
- Modified the experimental discussion to account for the additional baselines.
- Modified the statement of and discussion around Proposition 5.2.
- Added a section motivating the need for locality in the Appendix (Section C.4).
- Added an additional ablation motivating locality in the Appendix (Section C.4).
- Changed the experimental settings description in the Appendix (Section B).

---

### Meta-Review · Area_Chair_QK9w · 2023-12-08

**Metareview:**

This work proposes a novel graph-rewiring framework for Graph Neural Networks (GNNs) that addresses the issue of over-squashing and improves information flow by altering graph connectivity, while respecting the locality and preserving the sparsity of the graph, and demonstrates its effectiveness on several real-world benchmarks.

Strength
- This paper gives a good summarization of spectral and spatial rewiring methods. Upon which, a clear motivation Table 1 is given.
- The paper is overall well written and structured, theoretical justification is also developed to support the method.

Weakness
- Discussion with existing work is not sufficient.
- Lack comparison with two existing works that are frequently mentioned in the related works.

During the rebuttal and discussion phase, both two points in the weakness have been clarified.

**Justification For Why Not Higher Score:**

A new method to an old problem. The method itself may not be able to transfer to other problems, the scope of audience can be limited.

**Justification For Why Not Lower Score:**

Concerns have been addressed in the rebuttal and discussion.

---

### Decision · Program_Chairs · 2024-01-16

Accept (poster)